# Joint Graph Rewiring and Feature Denoising via Spectral Resonance

**Jonas Linkerhägner**\*  **Cheng Shi**\*  **Ivan Dokmanić**
Department of Mathematics and Computer Science
University of Basel
`firstname.lastname@unibas.ch`

## Abstract

When learning from graph data, the graph and the node features both give noisy information about the node labels. In this paper we propose an algorithm to **j**ointly **d**enoise the features and **r**ewire the graph (JDR), which improves the performance of downstream node classification graph neural nets (GNNs). JDR works by aligning the leading spectral spaces of graph and feature matrices. It approximately solves the associated non-convex optimization problem in a way that handles graphs with multiple classes and different levels of homophily or heterophily. We theoretically justify JDR in a stylized setting and show that it consistently outperforms existing rewiring methods on a wide range of synthetic and real-world node classification tasks.

## 1 Introduction

Graph neural networks (GNNs) are a powerful deep learning tool for graph-structured data, with applications in physics (Mandal et al., 2022; Linkerhägner et al., 2023), chemistry (Gilmer et al., 2017), biology (Gligorijević et al., 2021) and beyond (Zhou et al., 2020). Typical tasks across disciplines include graph classification (Duvenaud et al., 2015; Xu et al., 2019), node classification (Kipf and Welling, 2017; Li et al., 2019) and link prediction (Pan et al., 2022).

Graph datasets contain two distinct types of information: the graph structure and the node features. The graph encodes interactions between entities and thus the classes or communities they belong to, similarly to the features. Recent work demonstrates that *rewiring* the graph by judiciously adding and removing edges may improve downstream GNN performance. That work argues that in a GNN, the graph serves not only to encode interactions but also to organize message passing computations (Battaglia et al., 2018). Even when it correctly encodes interactions it may not be an effective *computational* graph—rewiring it may then facilitate information flow.

Graph rewiring methods can be categorized into preprocessing and end-to-end. Preprocessing methods rewire the graph by relating its geometric and spectral properties to information flow (Topping et al., 2022; Nguyen et al., 2023; Karhadkar et al., 2023). End-to-end methods (Giraldo et al., 2023; Gutteridge et al., 2023; Qian et al., 2024) dynamically rewire the graph during training, leveraging both the graph and the node features. Unlike preprocessing methods, they do not output an improved graph which restricts their interpretability and reusability. Our focus is on the preprocessing methods.

There are thus two mechanisms that hurt performance of GNNs: **(1)** real-world graphs and features are noisy (the graph has spurious and missing links), and **(2)** geometric properties of the graph impede message passing. In this paper we focus on **(1)** and ask a natural question: *can simple, joint feature and graph denoising improve performance of a downstream GNN?*

We propose a new rewiring scheme that also uses node features to produce an enhanced graph. We leverage the fact that both the graph and the features are correlated with the labels. This is explicit in high-quality stylized models of graphs with features, including community models such as the contextual stochastic block model (cSBM) (Deshpande et al., 2018) and neighborhood graphs on

---

\*These authors contributed equally.

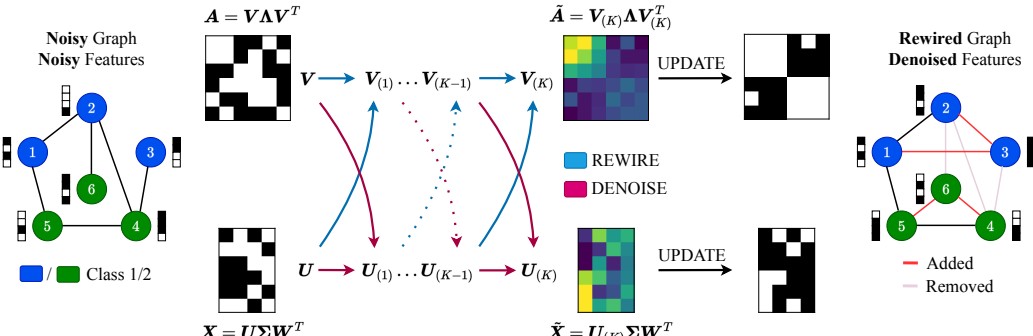

Figure 1: Schematic overview of joint denoising and rewiring (JDR). In this example, we consider a noisy graph as it occurs in many different real-world scenarios, in the sense that it contains edges between and within classes and its node features are not fully aligned with the labels. The graph's adjacency matrix $A$ and binary node features $X$ are decomposed via spectral decomposition and singular value decomposition (SVD). The rewiring of $A$ is performed by combining the information of its own eigenvectors $V$ and the singular vectors $U$ from $X$. The same applies vice versa for denoising, and both are performed iteratively $K$ times. We synthesize the rewired graph $\tilde{A}$ and the denoised features $\tilde{X}$ by multiplying back with the final $V_{(K)}$ and $U_{(K)}$. To get specific properties like sparsity or binarity we can perform an UPDATE step, e.g. by thresholding (as done here). The resulting denoised and rewired graph is displayed on the right. Its structure now better represents the communities and the first entry of the features indicates the class assignment.

points from low-dimensional manifolds. This fact motivates various spectral clustering and non-linear dimensionality reduction methods (Shi and Malik, 2000; Ng et al., 2001). In the cSBM, seminal theoretical work shows that jointly leveraging the graph (stochastic block model (SBM)) and the features (a Gaussian mixture model (GMM)) improves over unsupervised clustering using either piece of information alone. However, the associated efficient inference algorithms based on belief propagation (Deshpande et al., 2018; Duranthon and Zdeborová, 2023) rely on perfect knowledge of the distribution of the cSBM and cannot be applied to arbitrary real-world data.

Our contributions are as follows:

1. We take inspiration from work on the cSBM to design a practical algorithm for joint graph rewiring and feature denoising, which can improve the node classification performance of any downstream GNN on real-world data. We achieve this by adapting the graph and the features so as to maximize alignment between their leading eigenspaces. If these spaces are well-aligned we say that the graph and the features are *in resonance*.

2. We design an alternating optimization algorithm, joint denoising and rewiring (JDR), which approximates alignment maximization on spectrally-complex real-world graph data with multiple classes, possibly homophilic or heterophilic. We prove that JDR improves alignment between the graph and the features, but also with the labels, on a stylized generative model with noise from the Gaussian orthogonal ensemble (GOE); a recent conjecture in the literature suggests that this generalizes to cSBM.

3. We run extensive experiments to show that JDR outperforms existing preprocessing rewiring strategies while being guided solely by denoising.

This last point suggests that although there exist graphs with topological and geometrical characteristics which make existing rewiring schemes beneficial, a greater issue in real-world graphs is noise in the sense of missing and spurious links. This is true even when the graphs correctly reflect the ground truth information. In a citation network, for example, citations that *should* exist may be missing because of incomplete scholarship. Conversely, citations that *should not* exist may be present because the authors engaged in bibliographic ornamentation. Our method is outlined in Figure 1 and the code repository is available online[1].

---

[1] https://github.com/jlinki/JDR

## 2 JOINT DENOISING AND REWIRING

### 2.1 PRELIMINARIES

We let $\mathcal{G} = (\mathcal{V}, \mathcal{E})$ be an undirected graph with $|\mathcal{V}| = N$ nodes and an adjacency matrix $\boldsymbol{A}$. To each node we associate an $F$-dimensional feature vector and collect these vectors in the rows of matrix $\boldsymbol{X} \in \mathbb{R}^{N \times F}$. We make extensive use of the graph and feature spectra, namely the eigendecomposition $\boldsymbol{A} = \boldsymbol{V} \boldsymbol{\Lambda} \boldsymbol{V}^T$ and the SVD $\boldsymbol{X} = \boldsymbol{U} \boldsymbol{\Sigma} \boldsymbol{W}^T$, with eigen- and singular values ordered from largest to smallest. (As discussed below, in heterophilic graphs we order the eigenvalues of $\boldsymbol{A}$ according to their absolute value.) The graph Laplacian is $\boldsymbol{L} = \boldsymbol{D} - \boldsymbol{A}$, where $\boldsymbol{D}$ is the diagonal node degree matrix. For $k > 2$ node classes, we use one-hot labels $\boldsymbol{y} \in \{0, 1\}^{N \times k}$. We write $[L]$ for the set $\{1, 2, \ldots, L\}$. In the balanced two-class case, we consider nodes to be ordered so that the first half has label $\boldsymbol{y}_i = -1$ and the second half $\boldsymbol{y}_i = 1$. In semi-supervised node classification, the task is to label the nodes based on the graph ($\boldsymbol{A}$ and $\boldsymbol{X}$) and a subset of the labels $\boldsymbol{y}$. *Homophilic* graphs are those where nodes are more likely to connect with nodes with similar features or labels (e.g., friendship networks (McPherson et al., 2001)); *heterophilic* graphs are those where nodes more likely to connect with dissimilar nodes (e.g., protein interaction networks (Zhu et al., 2020)).

### 2.2 MOTIVATION VIA THE CONTEXTUAL STOCHASTIC BLOCK MODEL

For simplicity, we first explain our method for $k = 2$ classes and graphs generated from the cSBM. We then extend it to real-world graphs with multiple classes and describe the full practical algorithm.

**Contextual Stochastic Block Model.** CSBMs (Deshpande et al., 2018) extend SBMs (Abbe, 2018), a community graph model, by high-dimensional node features. They have become a key generative model for studying GNNs (Baranwal et al., 2021; Wu et al., 2023; Kothapalli et al., 2023); for further pointers see Appendix A.2. We use cSBMs to build intuition about the graph rewiring and denoising problem. In a balanced 2-class SBM, the nodes are divided into two equal-sized communities with node labels $\boldsymbol{y}_i \in \{\pm 1\}$. Pairs of nodes connect independently at random, with probability $c_{\text{in}}/N$ inside communities and $c_{\text{out}}/N$ across communities.

In the sparse regime (Abbe, 2018), with average node degree $d = \mathcal{O}(1)$, it is common to parameterize probabilities as $c_{\text{in}} = d + \lambda\sqrt{d}$ and $c_{\text{out}} = d - \lambda\sqrt{d}$, where $|\lambda|$ can be seen as the signal-to-noise ratio (SNR) of the graph. The signal $\boldsymbol{X}_i \in \mathbb{R}^F$ at node $i$ comes from a GMM,

$$\boldsymbol{X}_i = \sqrt{\frac{\mu}{N}} \boldsymbol{y}_i \boldsymbol{\xi} + \frac{\boldsymbol{z}_i}{\sqrt{F}}, \tag{1}$$

where $\boldsymbol{\xi} \sim \mathcal{N}(0, \boldsymbol{I}_F/F)$ is the randomly drawn mean and $\boldsymbol{z}_i \sim \mathcal{N}(0, \boldsymbol{I}_F)$ is i.i.d. Gaussian standard noise. We set $\gamma = \frac{N}{F}$ and, following Chien et al. (2021), parameterize the graphs generated from the cSBM using $\phi = \frac{2}{\pi} \arctan(\lambda\sqrt{\gamma}/\mu)$. For $\phi \to 1$ we get homophilic behavior; for $\phi \to -1$ we get heterophilic behavior. Close to either extreme the node features contain little information. For $\phi \to 0$ the graph is Erdős–Rényi and only the features contain information.

**Denoising and Rewiring the cSBM.** In the cSBM, $\boldsymbol{A}$ and $\boldsymbol{X}$ offer different noisy views on the labels. One can show that up to a scaling and a shift, the adjacency matrix is approximately $\pm\boldsymbol{y}\boldsymbol{y}^T + \boldsymbol{Z}_{ER}$, which means that it is approximately a rank-one matrix with labels in the range, corrupted with "Erdős–Rényi-like noise" $\boldsymbol{Z}_{ER}$ (Erdős and Rényi, 1959). Another way to see this is to note that $\mathbb{E}\boldsymbol{A} = \frac{1}{2N}(c_{\text{in}} + c_{\text{out}})\boldsymbol{1}\boldsymbol{1}^T + \frac{1}{2N}(c_{\text{in}} - c_{\text{out}})\boldsymbol{y}\boldsymbol{y}^T$ (from the definition of the SBM). Since $\boldsymbol{A}$ is close to $\mathbb{E}\boldsymbol{A}$ at high SNR, the eigenvectors contain information about the labels. It similarly follows directly from the definition that the feature matrix $\boldsymbol{X}$ is (up to a scaling) $\boldsymbol{y}\boldsymbol{u}^T + \boldsymbol{Z}_G$ where $\boldsymbol{Z}_G$ is white Gaussian noise. It thus makes sense to use the information from $\boldsymbol{X}$ to enhance $\boldsymbol{A}$ and vice versa. Deshpande et al. (2018) show that analyzing the following optimization problem:

$$\begin{aligned}
\underset{\boldsymbol{v} \in \mathbb{R}^N, \boldsymbol{u} \in \mathbb{R}^F}{\text{maximize}} \quad & \langle \boldsymbol{v}, \boldsymbol{A}\boldsymbol{v} \rangle + b \langle \boldsymbol{v}, \boldsymbol{X}\boldsymbol{u} \rangle \\
\text{subject to} \quad & \|\boldsymbol{v}\|_2 = \|\boldsymbol{u}\|_2 = 1, \langle \boldsymbol{v}, \boldsymbol{1} \rangle \leq \delta
\end{aligned} \tag{2}$$

for some carefully chosen value of $b$ allows one to characterize detection bounds in unsupervised community detection with $k = 2$. It is clear from the above reasoning that in the high-SNR regime ($\lambda$ and $\mu$ far away from the detection threshold), the second leading eigenvector of $\boldsymbol{A}$ and the leading

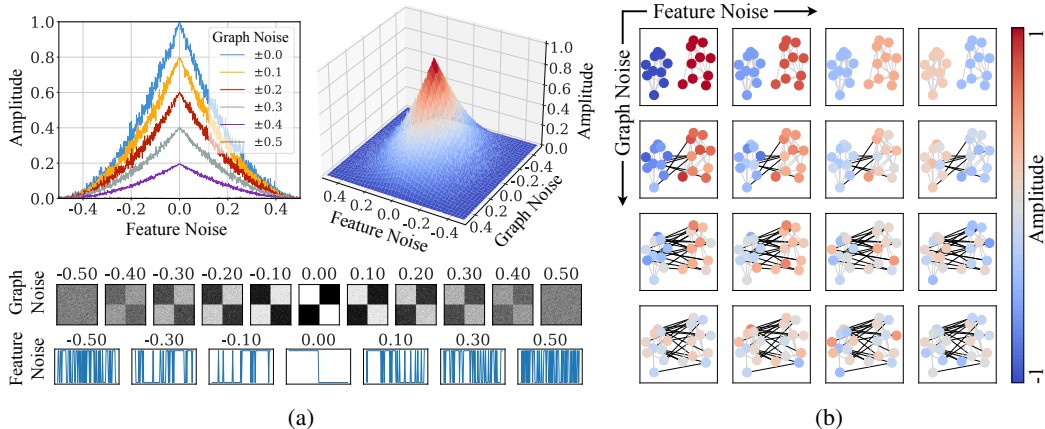

Figure 2: An illustration of spectral alignment and resonance. In (a) we plot $r = \boldsymbol{x}^T \boldsymbol{A} \boldsymbol{x}$ for different noise levels in $\boldsymbol{A}$ and $\boldsymbol{x} \in \{-1, 1\}^N$, illustrated in the rows below. Without noise, $\boldsymbol{x}$ is exactly the label vector and $\boldsymbol{A}$ is block-diagonal. We apply multiplicative noise; namely, for each noise level, we flip the sign of a proportion of values, resulting in a random signal for $\pm 0.5$. We see that the value of $r$ depends on the noise level. The maximum is achieved for zero noise when the second leading eigenvector of $\boldsymbol{A}$ and the signal $\boldsymbol{x}$ are perfectly aligned. In (b), we consider a signal $\hat{\boldsymbol{x}} = \boldsymbol{A} \boldsymbol{x}$ for different noise levels in $\boldsymbol{A}$ and $\boldsymbol{x}$ on a graph with 20 nodes; only a quarter of edges are shown to reduce clutter; the intra-class edges are grey; the inter-class edges are black. The largest norm is obtained for noise-free $\boldsymbol{A}$ and $\boldsymbol{x}$ (upper-left corner). The norm of $\hat{\boldsymbol{x}}$ and the separation of communities decrease along both noise axes. The inherent denoising capabilities of propagating $\boldsymbol{x}$ on a high-SNR graph (Ma et al., 2021b) are also visible, particularly in the first two rows to the right.

left singular vector of $\boldsymbol{X}$ approximately coincide with the labels. The optimal $\boldsymbol{v}^*$ is related to those vectors and aligned with the labels, since the quadratic and the bilinear form in (2) are individually maximized by the mentioned vectors. The maximizer of the linear combination of both terms therefore combines the spectral information from both matrices—the graph and the features. This suggests the following rationale for denoising: **(1)** We can interpret the value of (2) as a measure of alignment. Since $\boldsymbol{v}^*$ corresponds to the labels, we can relate this measure to the quality of the label estimation. **(2)** We may leverage this alignment to rewire the graph and denoise the features. Namely, we could perturb $\boldsymbol{A}$ and $\boldsymbol{X}$ in a way that improves the alignment.

In real datasets, however, the optimal value of $b$ is unknown, the scaling of $\boldsymbol{X}$ is arbitrary, and things are further complicated by having (many) more than 2 classes. Moreover, (2) is computationally hard. We thus define a simple related measure of alignment which alleviates these issues.

**Definition 1.** *Recall the decompositions $\boldsymbol{A} = \boldsymbol{V} \boldsymbol{\Lambda} \boldsymbol{V}^T$, $\boldsymbol{X} = \boldsymbol{U} \boldsymbol{\Sigma} \boldsymbol{W}^T$, and let $\boldsymbol{V}_L$, $\boldsymbol{U}_L$ denote the first $L$ columns of $\boldsymbol{V}$ and $\boldsymbol{U}$ and $\|.\|_{\mathrm{sp}}$ the spectral norm. We define graph–feature alignment as*

$$\mathrm{Alignment}_L(\boldsymbol{A}, \boldsymbol{X}) = \|\boldsymbol{V}_L^T \boldsymbol{U}_L\|_{\mathrm{sp}}. \tag{3}$$

*Remark:* The logic of this definition is that for a cSBM with high SNR and $k$ classes, the information about labels is indeed contained in the leading $L = k$ vectors of $\boldsymbol{V}$ and $\boldsymbol{U}$. This follows directly by generalizing the formulation in (2) to multiple classes and thus multiple eigenvectors (Decelle et al., 2011; Lesieur et al., 2017). The quantity $\mathrm{Alignment}_L(\boldsymbol{A}, \boldsymbol{X})$ is the cosine of the angle between the subspaces spanned by the columns of $\boldsymbol{V}_L$ and $\boldsymbol{U}_L$. To denoise the features and rewire the graph, we seek to improve the alignment.

Given $\mathrm{Alignment}_L(\boldsymbol{A}, \boldsymbol{X})$ and a graph with $\boldsymbol{A}_0$ and $\boldsymbol{X}_0$, the jointly denoised graph and features are the solution to

$$\underset{\boldsymbol{A}, \boldsymbol{X}}{\mathrm{maximize}}\, \mathrm{Alignment}_L(\boldsymbol{A}, \boldsymbol{X})$$
$$\text{subject to } \|\boldsymbol{A} - \boldsymbol{A}_0\| \leq \delta_A, \|\boldsymbol{X} - \boldsymbol{X}_0\| \leq \delta_X. \tag{4}$$

The parameters $\delta_A, \delta_X > 0$ modulate the strength of alignment. We will show empirically that a stronger alignment indicates a better representation of the labels by $\boldsymbol{A}$ and $\boldsymbol{X}$ and thus a better

graph. Figure 2 visualizes this connection. It shows that the response of the graph to features is maximized when the spectra of the graph and the features are aligned. We refer to the condition where the alignment is high as spectral resonance; see Appendix A.1.1 for further discussion.

## 2.3 JOINT DENOISING AND REWIRING ALGORITHM

Maximizing the alignment (4) directly, e.g., using gradient descent, is computationally challenging. Here we propose a heuristic which alternates between spectral interpolation and graph synthesis. We later prove that the resulting algorithm indeed improves alignment, both with the labels and between the graph and the features, under a stylized noise model. The algorithm, illustrated in Figure 1, comprises three steps. In Step 1, we compute the spectral decompositions of $A$ and $X$. To improve the alignment, we interpolate between the $L$ largest eigenvectors in Step 2. Based on the new eigenvectors, we synthesize a new graph in Step 3. The three steps are iterated until a stopping criterion is met. As is standard in the rewiring literature, the hyperparameters of the algorithm are tuned on a validation set. Formalizing this results in the JDR algorithm:

**Step 1:** Decomposition

$$A = V\Lambda V^T \text{ with } V = (v_1, v_2, \dots, v_N) \text{ and } X = U\Sigma W^T \text{ with } U = (u_1, u_2, \dots, u_N)$$

**Step 2:** Interpolation: For every $i \in [L]$,

$$\tilde{v}_i = (1 - \eta_A)v_i + \eta_A \operatorname{sign}(\langle v_i, u_j \rangle)u_j$$
$$\tilde{u}_i = (1 - \eta_X)u_i + \eta_X \operatorname{sign}(\langle u_i, v_j \rangle)v_j$$

where $j$ is chosen as $\operatorname{argmax}_{j \in [L]} |\langle v_i, u_j \rangle|$ when updating $v_i$ and as $\operatorname{argmax}_{j \in [L]} |\langle u_i, v_j \rangle|$ when updating $u_i$. $\eta_A$ and $\eta_X$ are hyperparameters that are tuned with a downstream algorithm on a validation set. We use $\operatorname{sign}()$ to handle sign ambiguities in decompositions.

**Step 3:** Graph Synthesis

$$\tilde{A} = \tilde{V}\Lambda\tilde{V}^T \text{ and } \tilde{X} = \tilde{U}\Sigma W^T$$

**Step 4:** Iterate steps $K$ times with

$$A \leftarrow \tilde{A} \text{ and } X \leftarrow \tilde{X}.$$

Following (3), we consider the $L$ leading eigenvectors of $A$ and $X$ for interpolation. Since these bases may be rotated with respect to each other (we note that (3) is insensitive to relative rotations, see Appendix A.1.2), when updating an eigenvector of $A$, we interpolate it with the most similar eigenvector of $X$. We show empirically that this heuristic yields strong results, but also prove that it improves alignment with labels with a stylized noise model. We emphasize that the interpolation rates $\eta_A$ and $\eta_X$ are the same across different eigenvectors and iterations $K$. After $K$ steps, we synthesize the final weighted dense graph $\tilde{A} = V_{(K)}\Lambda V_{(K)}^T$. To efficiently apply GNNs, we can enforce sparsity, e.g., via thresholding or selecting the top-$k$ entries per node. A detailed pseudocode is given in Appendix A.1.

**An illustration.** A simple edge case to illustrate how the algorithm works is when either only $A$ or $X$ contains information. In a cSBM with $\phi = 0$, $X$ contains all the information, so the best hyperparameter choice is $\eta_X = 0$ and (4) simplifies to a maximization over $A$. Since there are only two classes, it is sufficient to consider $L = 1$. From (2) we know that the leading left singular vector $u_1$ of $X$ is well-aligned with the labels. We thus replace the second leading eigenvector $v_2$ in $A$ by $u_1$ by choosing $\eta_A = 1.0$. After graph synthesis, the new $v_2$ of $\tilde{A}$ is not yet equal to $u_1$, since $u_1$ was not orthogonal to the other $v_i$. We thus repeat the three steps $K$ times. For $\phi = \pm 1$ all information is contained in the graph; a similar argument can then be constructed *mutatis mutandis*.

**JDR Improves Alignment.** We now show that JDR improves alignment, as defined in (4), under a stylized cSBM-like model. In fact, we show a stronger result: that the algorithm improves alignment with the true labels. Appealing to universality arguments (Hu and Lu, 2023), we study a model with a spiked Gaussian matrix $A^c = \frac{\lambda}{N}yy^T + \frac{1}{\sqrt{N}}O_A$ where $O_A$ is GOE noise instead of the binary matrix

Table 1: Comparison of state-of-the-art preprocessing rewiring approaches. Note that we refer to the computational complexity per iteration. $N$ denotes the number of nodes, $m$ the number of edges and $d_{\max}$ is the maximum node degree. Additional details on the complexity of JDR are given in Appendix A.1.3; detailed runtime comparisons are in Appendix A.1.4

| Method | Add edge | Remove edge | Use Features | Heterophilic? | Complexity |
|---|---|---|---|---|---|
| DIGL (Gasteiger et al., 2019) | ✓ | ✗ | ✗ | ✗ | $\mathcal{O}(N)$ |
| FoSR (Karhadkar et al., 2023) | ✓ | ✗ | ✗ | ✓ | $\mathcal{O}(N^2)$ |
| BORF (Nguyen et al., 2023) | ✓ | ✓ | ✗ | ✓ | $\mathcal{O}(md_{\max}^3)$ |
| **JDR (Ours)** | ✓ | ✓ | ✓ | ✓ | $\mathcal{O}(N)$ |

$\boldsymbol{A}$, and features $\boldsymbol{X} = \sqrt{\frac{\mu}{N}}\boldsymbol{y}\boldsymbol{\xi}^T + \frac{1}{\sqrt{F}}\boldsymbol{O}_X$ as defined in 1. In the cSBM context, a recent conjecture with strong empirical support states that replacing the binary $\boldsymbol{A}$ by $\boldsymbol{A}^c$ leads to the same behavior in downstream tasks such as community detection (Deshpande et al., 2018; Lu and Sen, 2023) and node classification (Shi et al., 2024). An iteration of JDR with $L = 1$ applied to this model, first interpolates between the leading eigenvector $\boldsymbol{v}_1(\boldsymbol{A}^c) = \boldsymbol{v}_A$ and leading left singular vector $\boldsymbol{u}_1(\boldsymbol{X}) = \boldsymbol{u}_X$. Graph and feature synthesis then yields $\boldsymbol{A}_{\eta_A}^c = \boldsymbol{A}^c + \lambda_1(\boldsymbol{A}^c)\left(-\boldsymbol{v}_A\boldsymbol{v}_A^T + \tilde{\boldsymbol{v}}_A\tilde{\boldsymbol{v}}_A^T\right)$ and $\boldsymbol{X}_{\eta_X} = \boldsymbol{X} + \sigma_1(\boldsymbol{X})\left(-\boldsymbol{u}_X\boldsymbol{w}_X^T + \tilde{\boldsymbol{u}}_X\boldsymbol{w}_X^T\right)$. Here $\tilde{\boldsymbol{v}}_A = (1 - \eta_A)\boldsymbol{v}_A + \operatorname{sign}(\langle\boldsymbol{v}_A, \boldsymbol{u}_X\rangle)\eta_A\boldsymbol{v}_X$ and $\tilde{\boldsymbol{u}}_X = (1 - \eta_X)\boldsymbol{u}_X + \operatorname{sign}(\langle\boldsymbol{v}_A, \boldsymbol{u}_X\rangle)\eta_X\boldsymbol{v}_A$, where $\boldsymbol{w}_1(\boldsymbol{X}) = \boldsymbol{w}_X$ is the leading right singular vector of $\boldsymbol{X}$. Denoting $\tilde{\boldsymbol{y}} = \boldsymbol{y}/\sqrt{N}$, we have

**Proposition 1.** *Let* $\lambda > 1$ *and* $\mu > \sqrt{\gamma}$ *with* $\gamma = N/F$. *There exist* $\eta_A^0, \eta_X^0 \in (0, 1)$ *such that for all* $\eta_A \in (0, \eta_A^0)$ *and* $\eta_X \in (0, \eta_X^0)$, *when* $N \to \infty$, *we have*

$$\left\langle\boldsymbol{v}_1(\boldsymbol{A}_{\eta_A}^c), \tilde{\boldsymbol{y}}\right\rangle^2 \overset{a.s}{>} \left\langle\boldsymbol{v}_1(\boldsymbol{A}^c), \tilde{\boldsymbol{y}}\right\rangle^2 \quad and \quad \left\langle\boldsymbol{u}_1(\boldsymbol{X}_{\eta_X}), \tilde{\boldsymbol{y}}\right\rangle^2 \overset{a.s}{>} \left\langle\boldsymbol{u}_1(\boldsymbol{X}), \tilde{\boldsymbol{y}}\right\rangle^2.$$

In words, interpolation improves alignment of the largest eigenvector with the labels $\boldsymbol{y}$ for sufficiently large graphs. The proof, based on the BBP transition in the spiked covariance model (Baik et al., 2005) and the fluctuation of the leading eigenvector, can be found in Appendix A.3. It seems challenging but quite possible to extend this argument to a binary $\boldsymbol{A}$. One would then interpolate between $\boldsymbol{u}_x$ and $\boldsymbol{A}$'s second leading eigenvector $\boldsymbol{v}_2(\boldsymbol{A})$, which has similar properties to $\boldsymbol{v}_1(\boldsymbol{A}^c)$, especially in a dense graph regime (Nadakuditi and Newman, 2012).

## 3  EXPERIMENTS

We extensively evaluate JDR on both synthetic data generated from the cSBM and real-world benchmark datasets. We follow experimental setting from Chien et al. (2021) and evaluate JDR for semi-supervised node classification with different downstream GNNs. We also adopt their data splits, namely the sparse splitting $2.5\%/2.5\%/95\%$ for training, validation and testing, respectively, or the dense splitting $60\%/20\%/20\%$. For the general experiments, we perform 100 runs with different random splits. For the scalability experiments, we use the experimental settings of the respective works (Lim et al., 2021; Platonov et al., 2023). We report the average accuracy and the $95\%$-confidence interval calculated via bootstrapping with 1000 samples. All experiments are reproducible using the code provided.

**Baselines.** Following recent works on rewiring, we use graph convolution network (GCN) (Kipf and Welling, 2017) as our downstream GNN. To obtain a more comprehensive picture, we additionally evaluate the performance on the more recent generalized PageRank graph neural network (GPRGNN) (Chien et al., 2021). We compare our algorithm with the state-of-the-art rewiring methods first-order spectral rewiring (FoSR) (Karhadkar et al., 2023), batch Ollivier-Ricci flow (BORF) (Nguyen et al., 2023) and diffusion improves graph learning (DIGL) (Gasteiger et al., 2019). FoSR approximates which edges should be added to maximize the spectral gap to reduce oversquashing. BORF adds edges in regions of negative curvature in the graph, which indicate bottlenecks that can lead to an oversquashing of the messages passed along these edges. A positive curvature indicates that there are so many edges in this area that messages could be oversmoothed, which is why edges are removed here. We compare computational and implementation aspects of JDR and baselines in Table 1. On the cSBM, we compare to an *optimal* algorithm, namely the approximate message passing-belief propagation (AMP-BP) algorithm (Duranthon and Zdeborová, 2023). AMP-BP

is asymptotically optimal (in the large dimension limit) for unsupervised or semi-supervised community detection in the cSBM. It relies on knowing the distribution of the cSBM and is thus not applicable to real-world graphs with unknown characteristics and complex features.

**Hyperparameters.** Unless stated otherwise, we use the hyperparameters from Chien et al. (2021) for the GNNs and optimize the hyperparameters of JDR using a mixture of grid and random search on the validation set. We use the top-64 values of $\tilde{A}$ to enforce sparsity and interpolation to update the features. For DIGL, FoSR and BORF, we tune their hyperparameters using a grid search, closely following the given parameter range from the original papers. For all hyperparameter searches we use GCN and GPRGNN as the downstream models on 10 runs with different random splits. A detailed list of all hyperparameters can be found in Appendix A.7 or in the code repository.

## 3.1 RESULTS ON SYNTHETIC DATA

We first test JDR on data generated from the cSBM, as we can easily vary the SNR of the graph and the features to verify its denoising and rewiring capabilities. We focus on the sparse splitting, since for the dense splitting GPRGNN already matches the performance of AMP-BP.

**Does JDR Maximize Alignment?** Before discussing Figure 4, which shows the results of baselines and JDR for different values of $\phi$, we verify empirically that our alternating optimization algorithm indeed approximates solutions to (4). As shown in Figure 3, the quantity $\mathrm{Alignment}_L(A, X)$ improves significantly after running JDR, across all $\phi$. As we show next, this happens simultaneously with improvements in downstream performance, which lends credence to the intuitive reasoning that motivates our algorithm. For additional alignment results on real-world data and baselines, refer to Appendix A.5.5.

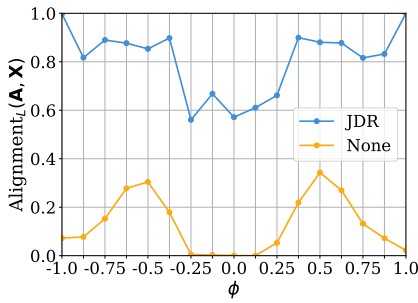

Figure 3: Alignment of the leading eigenspaces according to (3) for graphs from the cSBM with different $\phi$.

**Heterophilic Regime.** For $\phi < -0.25$, the predictions of GCN are only slightly better than random. GPRGNN performs much better, since it can learn higher order polynomial filters to deal with heterophily. GCN+JDR outperforms the baseline by a very large margin; it handles heterophilic data well. Using JDR for GPRGNN further improves its already strong performance in this regime. Both GNNs benefit less from the denoising in the weakly heterophilic setting where they exhibit the worst performance across all $\phi$. The difference between $\phi = 0$ and the weakly heterophilic regime is that "optimal denoising" for $\phi = 0$ is straightforward, since all the information is contained in $X$. We show similar findings for spectral clustering on the cSBM in Appendix A.5.6.

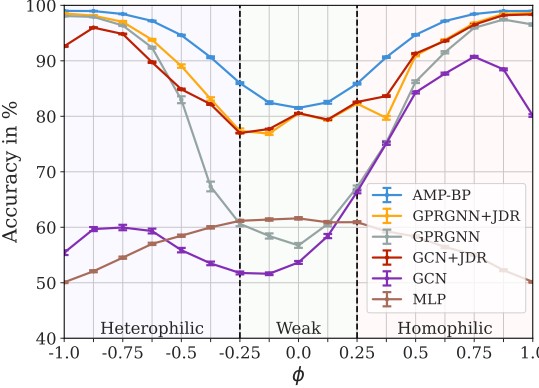

Figure 4: Test accuracy on graphs from the cSBM across different $\phi$. The error bars indicate the 95% confidence interval. JDR improves the performance for both GNNs across all $\phi$.

**Weak Graph Regime.** For $|\phi| \leq 0.25$, where the SNR of the graph is very low, both GNNs perform poorly. Intuitively, when the graph is very noisy, a GNN is a suboptimal model, since it leverages the graph structure. A simple MLP baseline, using only the node features, outperforms GNNs in this setting, with all three approaches lagging far behind AMP-BP. Using JDR, we see significant improvements for both GNNs, which almost catch up with AMP-BP for $\phi = 0$. Although all information was available in the node features, the GNN with JDR now clearly outperform the MLP by a very large margin. We argue that this is because in the semi-supervised setting with few labels available, the GNN generalizes much better.

**Homophilic Regime.** For $\phi > 0.25$, GCN and GPRGNN perform similarly well, with

Table 2: Results on real-world homophilic datasets in the sparse splitting (2.5%/2.5%/95%): Mean accuracy across runs (%) ± 95% confidence interval. Best average accuracy in **bold**.

| Method | Cora | CiteSeer | PubMed | Computers | Photo |
|---|---|---|---|---|---|
| GCN | 77.26±0.35 | 67.16±0.37 | 84.22±0.09 | 84.42±0.31 | 91.33±0.29 |
| GCN+DIGL | 79.27±0.26 | 68.03±0.33 | 84.60±0.09 | **86.00±0.24** | 92.00±0.23 |
| GCN+FoSR | 77.23±0.34 | 67.03±0.34 | 84.21±0.09 | 84.34±0.27 | 91.36±0.28 |
| GCN+BORF | 77.23±0.35 | 66.96±0.38 | 84.22±0.09 | 84.46±0.30 | 91.26±0.30 |
| GCN+JDR | **79.96±0.26** | **69.35±0.28** | **84.79±0.08** | 85.66±0.36 | **92.52±0.23** |
| GPRGNN | 79.65±0.33 | 67.50±0.35 | 84.33±0.10 | 84.06±0.48 | 92.01±0.41 |
| GPRGNN+DIGL | 79.77±0.30 | 67.50±0.35 | 84.72±0.10 | **86.25±0.28** | 92.31±0.25 |
| GPRGNN+FoSR | 79.22±0.31 | 67.30±0.38 | 84.32±0.09 | 84.21±0.46 | 92.07±0.37 |
| GPRGNN+BORF | 79.43±0.30 | 67.48±0.36 | 84.36±0.10 | 84.08±0.43 | 92.11±0.38 |
| GPRGNN+JDR | **80.77±0.29** | **69.17±0.30** | **85.05±0.08** | 84.77±0.35 | **92.68±0.25** |

Table 3: Results on real-world heterophilic dataset in the dense splitting (60%/20%/20%): Mean accuracy across runs (%) ± 95% confidence interval. Best average accuracy in **bold**.

| Method | Chameleon | Squirrel | Actor | Texas | Cornell |
|---|---|---|---|---|---|
| GCN | 67.65±0.42 | 57.94±0.31 | 34.00±0.31 | 75.62±1.12 | 64.68±1.25 |
| GCN+DIGL | 58.04±0.48 | 39.64±0.34 | 39.57±0.29 | **91.05±0.73** | **88.49±0.74** |
| GCN+FoSR | 67.67±0.39 | 58.12±0.35 | 33.98±0.30 | 78.31±1.07 | 65.64±1.06 |
| GCN+BORF | 67.78±0.43 | OOM | 33.95±0.31 | 76.66±1.10 | 68.72±1.11 |
| GCN+JDR | **69.76±0.50** | **61.76±0.39** | **40.47±0.31** | 85.12±0.74 | 84.51±1.06 |
| GPRGNN | 69.15±0.51 | 53.44±0.37 | 39.52±0.22 | 92.82±0.67 | 87.79±0.89 |
| GPRGNN+DIGL | 66.57±0.46 | 42.98±0.37 | 39.61±0.21 | 91.11±0.72 | 88.06±0.81 |
| GPRGNN+FoSR | 68.96±0.45 | 52.34±0.37 | 39.47±0.21 | 93.16±0.66 | 87.51±1.04 |
| GPRGNN+BORF | 69.44±0.56 | OOM | 39.55±0.20 | 93.53±0.68 | 88.83±1.06 |
| GPRGNN+JDR | **71.00±0.50** | **60.62±0.38** | **41.89±0.24** | **93.85±0.54** | **89.45±0.84** |

GPRGNN achieving better results for $\phi \to 1.0$. With JDR, they become much more comparable to each other and closer to AMP-BP. Even though the hyperparameters of JDR were tuned using only GCN as a downstream model, it also improves the performance of GPRGNN for all $\phi$. The general robustness to hyperparameter changes is also analyzed in detail in Appendix A.6.

## 3.2 RESULTS ON REAL-WORLD DATA

We evaluate JDR on five common homophilic benchmarks datasets, namely the citation graphs Cora, CiteSeer, PubMed (Sen et al., 2008) and the Amazon co-purchase graphs Computers and Photo (McAuley et al., 2015). For heterophilic datasets, we rely on the Wikipedia graphs Chameleon and Squirrel (Rozemberczki et al., 2021), the WebKB datasets Texas and Cornell used in Pei et al. (2020) and the actor co-occurence network Actor (Tang et al., 2009). To show the scalability of JDR on larger heterophilic datasets, we further report the results for the Yandex Q user network Questions (Platonov et al., 2023) and the social networks Penn94 and Twitch-Gamers (Lim et al., 2021). Further details about all datasets are in Appendix A.4. Following Chien et al. (2021), we evaluate the homophilic datasets in the sparse splitting, staying close to the original setting of Kipf and Welling (2017) and the heterophilic datasets in dense splitting (Pei et al., 2020). The remaining larger graphs are evaluated using their original splits. For further results and splits, see Appendix A.5.

**Homophilic Datasets.** Table 2 shows the results of JDR compared to the baselines. For both GNNs, JDR achieves the best results on four out of five datasets. GCN and GPRGNN with JDR achieve similar performance here, which is consistent with the findings for the homophilic cSBM. DIGL also performs strongly on the datasets and ranks first on Computers. However, with GPRGNN as a downstream model, the improvements are quite small. FoSR and BORF only marginally improve the performance of the GNNs in this setting.

**Heterophilic Datasets.** The results in Table 3 show that GCN+JDR can catch up significantly compared to GPRGNN, but GPRGNN+JDR generally performs better. This is in line with the findings for the heterophilic cSBM. DIGL performs well on Actor, Texas and Cornell despite its inherent homophily assumption. The reason for this is the chosen smoothing kernel, which results in a graph that is evenly connected everywhere with small weights. GCN then largely ignores the graph and thus performs very similarly to an MLP, which performs already quite well on these datasets (Chien et al., 2021). However, this fails for GPRGNN, which can make better use of the weak, complex graph structures. FoSR and BORF also improve performance here in most cases, but they are outperformed by JDR in all cases, often by a large margin. The out-of-memory error on Squirrel for BORF results from its computational complexity of $\mathcal{O}(md_{\max}^3)$, because the dataset has a large number of edges $m$ and a high maximum node degree $d_{\max}$.

**Larger Graphs.** Scalability is a problem for preprocessing rewiring methods because applying them to large graphs requires significant amounts of memory and compute (see complexity in Table 1). Since the decompositions needed for JDR can be truncated to the largest $L$ vectors, it is still applicable to larger graphs. The experimental results on larger heterophilic datasets in Table 4 verify this. They show that JDR can significantly improve performance for these larger

Table 4: Results on large datasets. Mean accuracy (ROC AUC for imbalanced Questions) across runs (%) $\pm 95\%$ confidence interval. Best results in **bold**.

| Method | Questions | Penn94 | Twitch-gamers |
|---|---|---|---|
| # nodes | $48,921$ | $41,554$ | $168,114$ |
| # edges | 0.15M | 1.36M | 6.8M |
| GCN | $75.31 \pm 0.81$ | $80.40 \pm 0.18$ | $64.56 \pm 0.19$ |
| GCN+DIGL | $73.35 \pm 0.64$ | $74.70 \pm 0.32$ | $61.64 \pm 0.14$ |
| GCN+FoSR | $75.51 \pm 0.73$ | $80.54 \pm 0.31$ | $64.65 \pm 0.15$ |
| GCN+JDR | $\mathbf{77.52 \pm 0.63}$ | $\mathbf{82.30 \pm 0.61}$ | $\mathbf{65.14 \pm 0.19}$ |

graphs, while FoSR only achieves marginal improvements. BORF ran out of memory on all of these datasets and DIGL is unable to improve due to its inherent homophily assumption. While scaling JDR to even larger graphs with millions of nodes is possible in principle, it requires more optimized and efficient implementations and is therefore left for future work.

## 4 RELATION TO PRIOR WORK

JDR is most related to preprocessing rewiring methods which we thus use as baselines. To provide a more thorough overview, we also place it within the extended literature.

**Graph Rewiring.** Recent work show that even when a graph correctly encodes interactions it may not be an effective *computational* graph for a GNN due to conditions such as *oversquashing* (Alon and Yahav, 2021; Di Giovanni et al., 2023) and *oversmoothing* (Chen et al., 2020a). Recently many methods have been proposed to address this, notably *graph rewiring methods*. They can be divided into preprocessing and end-to-end methods. Preprocessing methods rewire the graph using geometric and spectral properties, including curvature (Topping et al., 2022; Nguyen et al., 2023; Fesser and Weber, 2024; Bober et al., 2024), expansion (Deac et al., 2022; Banerjee et al., 2022), effective resistance (Black et al., 2023; Shen et al., 2024), and spectral gap (Karhadkar et al., 2023). Conceptually related is diffusion-based rewiring (Gasteiger et al., 2019) that smooths the graph with a diffusion kernel. This can be interpreted as graph denoising, but is only suitable for homophilic graphs. Our approach is related to rewiring but with several key differences (see Table 1). Our rewiring strategy aims to denoise the graph (rather than control some geometric property) with the goal to improve downstream performance, while the classical rewiring literature focuses on optimizing the graph for message passing computations.

Early end-to-end methods randomly drop edges during training to reduce oversmoothing (Rong et al., 2020). Subsequent work (Gutteridge et al., 2023; Qian et al., 2024) incorporates latent features to dynamically rewire the graph. Ji et al. (2023) use the estimated labels of a GNN to rewire the graph during training of the same GNN. Giraldo et al. (2023) use curvature information for dynamic rewiring. Graph Transformers (Dwivedi and Bresson, 2021; Rampasek et al., 2022) aim to overcome oversquashing in GNNs via global attention. In order to handle large graphs, these works still need to revert to sparse, non-global attention (Gabrielsson et al., 2023; Shirzad et al., 2023). Unlike preprocessing methods, end-to-end methods cannot output an improved graph which restricts their interpretability and reusability.

**Graph Denoising.** There is extensive literature on denoising signals on graphs using graph filters (Chen et al., 2014; Ma et al., 2021b; Liu et al., 2022). However, we are interested in modifying the structure of the graph itself (rewiring), in a way that can benefit any downstream algorithm. Dong and Kluger (2023) recently proposed a new metric to measure graph noise that correlates well with GCN performance. Based on this, they develop a method for graph denoising via self-supervised learning and link prediction. We discuss the relation to our work in detail in Appendix A.5.7 and also evaluate our rewired graphs using their ESNR metric there. More broadly, link prediction (Zhang and Chen, 2018; Pan et al., 2022) can be seen as a tool for graph denoising; this perspective has been applied, for instance, to denoising neighborhood graphs arising in molecular imaging (Debarnot et al., 2022).

**Graph Structure Learning.** The aim of graph structure learning (GSL) (Zhu et al., 2022) is to make GNNs more robust against adversarial perturbations of the graph or to learn a graph for data where there is no graph to start with (Jin et al., 2020; Chen et al., 2020b; Wang et al., 2024; Zhu et al., 2024). Lv et al. (2023) build a neighborhood graph over features and interpolate between it and the input graph, which is a form of alignment. Unlike our method, they do not use spectral information, are unable to deal with noisy features and are only suitable for homophilic graphs where similarity-based connection rules apply. Even though both our work and GSL consider noisy graph settings, they are conceptually very different. We do not add noise to graph datasets which corresponds to a perturbation rate of 0 ("clean data") in GSL nomenclature. Instead, we acknowledge that in every real world dataset, there is noise in the graph structure and the node features, and one manifestation of this noise is a misalignment of their leading eigenspaces. We then use this to rewire the graph (and denoise features) so as to improve the overall node classification performance of downstream GNNs. Naturally, GSL methods have difficulties to improve over baselines in this setting (Jin et al., 2020; Dong and Kluger, 2023). Our method, on the other hand, is not designed to handle strong perturbations and therefore cannot compete with GSL methods developed for specifically this purpose.

**Graph Regularization.** Laplacian regularization (Ando and Zhang, 2006), originally stemming from semi-supervised representation learning, has been adapted by recent methods (Yang et al., 2021; Ma et al., 2021a) to also improve the performance of GNNs. An extra loss term is added during the GNNs training, which contains additional information about the graph structure to reduce oversmoothing. Their main limiting factor is the underlying homophiliy assumption: It is assumed that connected nodes are more likely to share the same label.

## 5    Conclusion and Limitations

Our experimental results clearly show that spectral resonance is a powerful principle on which to build graph rewiring (and feature denoising) algorithms. JDR consistently outperforms existing rewiring methods DIGL, FoSR and BORF on both synthetic and real-world graph datasets. The smaller performance gains of GPRGNN suggest that this more powerful GNN is already able to leverage the complementary spectra of graphs and features to some extent.

The main limitation of JDR is that it cannot be used without node features. The preprocessing rewiring methods that we compare with do not have this limitation as they only use the graph for rewiring, but in turn, they cannot take advantage of features. Since JDR is the first method to jointly denoise the graph and the features, there are no other methods to which it could be directly compared. Our experiments thus highlight what advantage features bring to rewiring.

Furthermore, our results suggest that noise in real-world graphs is an important limiting factor for the performance of GNNs. It would be interesting to see whether feature-agnostic rewiring from a denoising perspective, for example using link prediction, could be used to improve the downstream performance. A related idea that we tested but could not get to work well is to combine existing geometric rewiring algorithms with JDR. Intuitively, there should be a way to benefit from both removing noise and facilitating computation, but we have to leave that exploration for future work.

We also note that most current rewiring methods can be applied to graph level tasks, while JDR is currently limited to node classification. It is an open question how to extend the cSBM idea to graph-level problems.

ACKNOWLEDGMENTS

JL, CS and ID were supported by the European Research Council (ERC) Starting Grant 852821—SWING.

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

# A APPENDIX

## A.1 THE JDR ALGORITHM

---

**Algorithm 1** Joint Denoising and Rewiring

---

1: **procedure** REWIRE($X$, $A$)           ▷ For DENOISE just exchange $X$ and $A$
2:      $X = U\Sigma W^T$
3:      $A = V\Lambda V^T$
4:      **for** $i$ **in** range($L_A$) **do**           ▷ Loop over $L_A$ eigenvectors in $A$
5:          $v_a \leftarrow V[:, i]$
6:          **for** $j$ **in** range($L_A$) **do**        ▷ Loop over $L_A$ eigenvectors in $X$
7:              $u_x \leftarrow U[:, j]$
8:              $\theta \leftarrow \langle u_x, v_a \rangle$        ▷ Find angle between eigenvectors
9:              **if** $|\theta| > |\theta_{\max}|$ **then**
10:                 $\theta_{\max} \leftarrow \theta$
11:                 $u_x^{\max} \leftarrow u_x$
12:              **end if**
13:          **end for**
14:          $\tilde{V}[:, i] \leftarrow (1 - \eta_A)v_a + \eta_A \text{sign}(\theta_{\max})u_x^{\max}$    ▷ Interpolation between eigenvectors
15:      **end for**
16:      $\tilde{A} \leftarrow \tilde{V}\Lambda\tilde{V}^T$
17: **end procedure**
18: $\tilde{X}, \tilde{A} \leftarrow X, A$
19: **for** $i$ **in** range(K) **do**                        ▷ Main loop
20:      $X' \leftarrow$ DENOISE($\tilde{X}, \tilde{A}$)
21:      $A' \leftarrow$ REWIRE($\tilde{X}, \tilde{A}$)
22:      $\tilde{X}, \tilde{A} \leftarrow X', A'$
23: **end for**
24: $\tilde{X} = $ UPDATE_X($X, \tilde{X}$)          ▷ Sparsify and binarize if needed
25: $\tilde{A} = $ UPDATE_A($A, \tilde{A}$)

---

### A.1.1 LOW-DIMENSIONAL GRAPHS AND RELATION TO RESONANCE

low-dimensional coordinates, We finally mention that although our algorithm is motivated by the cSBM, it could have equivalently been motivated by ubiquitous low-dimensional graphs. In such graphs, node labels are related to the which are in turn given by the eigenvectors of the graph Laplacian; this is illustrated in Figure 5. If, for example, the labels are given by the sign of the first non-constant eigenfunction (the slowest-changing normal mode), our notion of alignment with $L = 1$ clearly remains meaningful.

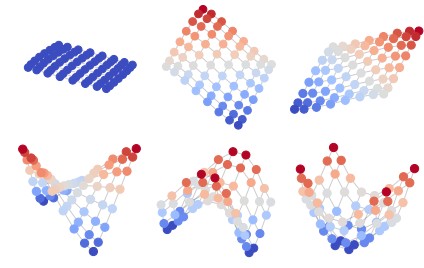

This also further motivates our terminology of resonance. In a purist sense, resonance is a dynamical phenomenon where driving a system with a frequency corresponding

Figure 5: Visualization of the first six eigenmodes of $L$ of the $8 \times 8$ grid graph.

to an eigenvalue of the Laplacian yields a diverging response. Importantly, the shape of the response is then an eigenfunction. In a broad sense, resonance signifies alignment with Laplacian eigenfunctions, which are the natural modes. For graphs, this is closely related to alignment with eigenvectors of the adjacency matrix (it is equivalent for $d$-regular graphs). As Figure 2b shows, maximizing alignment between feature and graph spectra indeed leads to the largest response of the graph to the features.

### A.1.2 ROTATIONAL INVARIANCE OF ALIGNMENT

We show that the alignment measure (3) is invariant to rotations of the subspaces for non-unique eigenvalues. Let $A \in \mathbb{R}^{N \times N}$ be the adjacency matrix and $X \in \mathbb{R}^{N \times F}$ the feature matrix. Let the eigendecomposition of $A$ be

$$A = \lambda_1 U_1 U_1^T + \cdots + \lambda_p U_p U_p^T$$

where $p \leq N$, $U_i \in \mathbb{R}^{N \times s_i}$ with $s_i$ being the multiplicity of the eigenvalue $\lambda_i$, $\sum_i s_i = N$, $U_i^T U_i = I$, and $U_j^T U_i = 0$ for $i \neq j$. Let similarly the SVD of $X$ be

$$X = \sigma_1 V_1 W_1^T + \cdots + \sigma_q V_q W_q^T,$$

where $V_i \in \mathbb{R}^{N \times t_i}$, $W_i \in \mathbb{R}^{F \times t_i}$, $\sum_i t_i = F$, with analogous orthogonality conditions. Assume both $(\lambda_i)$ and $(\sigma_i)$ are sorted from largest to smallest. Assume also for simplicity that $L = \sum_{i=1}^{p'} s_i = \sum_{i=1}^{q'} t_i$ so that the leading $L$-dimensional subspace of the graph $A$ is spanned by the columns of the block matrix $U_L = [U_1 \cdots U_{p'}]$. Of course it is also spanned by the columns of $\widetilde{U}_L = [U_1 Q_1 \cdots U_{p'} Q_{p'}]$ where invertible matrices $Q_i \in \mathbb{R}^{s_i \times s_i}$ reflect the fact that the eigensolver may return any of the infinitely many (when $s_i > 1$) orthogonal bases for the subspaces spanned by the columns of $U_i$. The $Q_i$ are orthogonal since $U_i Q_i$ are orthogonal. Similarly the leading $L$-dimensional subspace of the features is spanned by the columns of $V_L = [V_1 \cdots V_{q'}]$ but also of $\widetilde{V}_L = [V_1 R_1 \cdots V_{q'} R_{q'}]$ for any orthogonal $(R_i)$. Now

$$\|\widetilde{U}_L^T \widetilde{V}_L\|_{\mathrm{sp}} = \|\mathrm{blockdiag}(Q_1^T, \ldots, Q_{p'}^T)\, U_L^T V_L\, \mathrm{blockdiag}(R_1, \cdots, R_{q'})\|_{\mathrm{sp}}$$
$$= \|U_L^T V_L\|_{\mathrm{sp}}$$

for any choice of $Q$s and $R$s since both block-diagonal matrices are orthogonal and the spectral norm is unitarily-invariant.

### A.1.3 COMPUTATIONAL COMPLEXITY

The complexity of JDR results mainly from the SVD and eigendecomposition, which is of order $\mathcal{O}(FN\min(F, N))$ for SVD and $\mathcal{O}(N^3)$ for the eigendecomposition ($F = N$). Since we only need the leading $k$ eigenvectors this reduces to $\mathcal{O}(FNk)$. If the matrix is additionally sparse as is often the case for real-world graphs with binary node features this reduces further to $\mathcal{O}(\mathrm{nnz}(A)k)$ where $\mathrm{nnz}(A)$ is the number of non-zero elements in $A$. Since usually neither the average degree $d$ nor $k$ is scaled by $N$, the complexity actually scales with $\mathcal{O}(N)$.

### A.1.4 RUNTIME COMPARISON

In addition to computational complexity, we report measurements of running time of the different algorithms. We run JDR and baseline methods on the real-world datasets, using GCN as downstream model. All algorithms are run on Nvidia A100 with 80GB and we time their Python processes. We emphasize that we did not explicitly optimize the timing code and we kept the outputs and logging turned on. But this influences all methods in the same way so the relative comparisons are meaningful. The results in Table 5 do not show a clear "winner". The ambiguity is especially visible on the large heterophilic graphs, where JDR is slower than DIGL, but significantly faster than FoSR on two datasets. On the Twitch-gamers dataset, on the other hand, it is faster than DIGL but not as fast as FoSR. The main reason for this is that different hyperparameters choices of the rewiring methods lead to dramatically different run times (even when applying the same method on the same dataset). For example on Computers, JDR is very fast since it only requires 3 denoising iterations, compared to 15 on Citeseer. The same holds true for FoSR which only require 5 iterations on Twitch-gamers, but 700 on Questions. So if one wants to optimize for speed, one should constrain the hyperparameters of the methods that significantly impact execution speed. Of course, there is a trade-off between any such constraint and accuracy, as our experiments on the denoising iterations of JDR in Figure 16 (b) and Figure 17 (b) in Appendix A.6 show.

## A.2 CONTEXTUAL STOCHASTIC BLOCK MODELS

SBMs and GMMs are landmark theoretical models for studying clustering, classification problems and developing algorithmic tools. The cSBM (Deshpande et al., 2018), a combination of the two,

Table 5: Timing experiments **in seconds** for different rewiring methods using GCN as downstream GNN. Smaller is better. We record the time of the preprocessing and training and evaluating the GNN on 100 random splits. The results do not show a clear winner; JDR generally requires a comparable or less time compared to baselines. For more discussion see A.1.4.

| **Dataset** | Base | DIGL | FoSR | BORF | JDR |
|---|---|---|---|---|---|
| Cora | 182 | 228 | 187 | 201 | 246 |
| Citeseer | 258 | 360 | 291 | 290 | 433 |
| PubMed | 291 | 692 | 416 | 897 | 858 |
| Computers | 274 | 444 | 516 | 718 | 465 |
| Photo | 213 | 299 | 220 | 801 | 330 |
| Chameleon | 372 | 239 | 396 | 483 | 545 |
| Squirrel | 1263 | 302 | 1282 | - | 1659 |
| Actor | 166 | 286 | 171 | 225 | 319 |
| Texas | 163 | 200 | 169 | 174 | 203 |
| Cornell | 167 | 202 | 164 | 240 | 208 |
| Questions | 93 | 804 | 11053 | - | 3707 |
| Penn94 | 164 | 232 | 3198 | - | 1779 |
| Twitch-gamers | 1579 | 6729 | 1618 | - | 5165 |

has become a key model for studying node classification problems on graphs, inspiring numerous designs of GNNs like GPRGNN (Chien et al., 2021), GIANT (Chien et al., 2022) or ASGC (Chanpuriya and Musco, 2022). Many theoretical studies of node-level GNN problems are based on the cSBM, e.g. on double descent (Shi et al., 2024), neural collapse (Kothapalli et al., 2023), OOD generalization (Baranwal et al., 2021), or oversmoothing (Wu et al., 2023). Beyond being a standard synthetic benchmark, the cSBM is also used to to verify hypotheses about GNNs (Ma et al., 2022; Luan et al., 2023).

As for any model, the cSBM also comes with limitations. One possible limitation of our work is that cSBM assumes that the features are linear as in a GMM, which makes a linear classifier optimal. If the class boundaries are highly nonlinear, this is no longer true, and the spectrum of $X$ may need to be "linearized", e.g. via Laplacian eigenmaps or diffusion maps. Still, the results on real-world data show that the cSBM model is already highly transferable, suggesting that the high-dimensional features in real-world graph datasets are often quite linear.

## A.3 Proof of Proposition 1

**Notation.** We order the eigenvalues and singular values from largest to smallest and denote the eigenvector associated with the eigenvalue $\lambda_j$ by $v_j(A^c)$. For the leading eigenvalue and eigenvectors of $A^c$, we write $\lambda_1 = \lambda_A$ and $v_1(A^c) = v_A$. We use analogous notation for the singular values and corresponding singular vectors of $X$. For simplicity and without loss of generality we assume that the angles between these vectors are accute, i.e., $\langle v_A, u_X \rangle, \langle \tilde{y}, v_A \rangle, \langle \tilde{y}, u_X \rangle \geq 0$.

*Proof.* When $\lambda > 1$, based on the Baik-Ben Arous-Péché (BBP) transition (Baik et al., 2005; Paul, 2007), the leading eigenvalue of $A$ lies outside the spectral bulk,

$$\lambda_A = \lambda + \frac{1}{\lambda} + \mathcal{O}_p\left(\frac{1}{\sqrt{N}}\right),$$

and the fluctuation of the leading eigenvector satisfies

$$q_A := \lambda\left(v_A - \sqrt{1 - \frac{1}{\lambda^2}}\tilde{y}\right) \xrightarrow{d} \text{Haar}(\mathbb{S}_{\tilde{y}_\perp}^{N-2}) \tag{5}$$

where $\text{Haar}(\mathbb{S}_{\tilde{y}_\perp}^{N-2})$ is the uniform distribution on the sphere orthogonal to $\tilde{y}$, $\mathbb{S}_{\tilde{y}_\perp}^{n-2} = \{v : v \in \mathbb{R}^N \mid v^T\tilde{y} = 0, \|v\| = 1\}$, and the convergence is in distribution as $N \to \infty$. Similarly,

for the rectangular matrix $\boldsymbol{X}$ we have (Benaych-Georges and Nadakuditi, 2012),

$$\sigma_X = \sqrt{\frac{(\gamma + \mu)(1 + \mu)}{\mu}} + \mathcal{O}_p\left(\frac{1}{\sqrt{N}}\right)$$

and the fluctuation of the leading singular vector satisfies

$$\boldsymbol{q}_X := \sqrt{\frac{\mu(\mu + \gamma)}{\gamma(1 + \mu)}}\left(\boldsymbol{u}_X - \sqrt{1 - \frac{\gamma(1 + \mu)}{\mu(\mu + \gamma)}}\tilde{\boldsymbol{y}}\right) \xrightarrow{d} \text{Haar}(\mathbb{S}_{\tilde{\boldsymbol{y}}^{\perp}}^{N-2}).$$

To denoise $\boldsymbol{A}^c$, we adjust the leading eigenvector towards the direction of $\boldsymbol{u}_X$ as

$$\tilde{\boldsymbol{v}}_A = (1 - \eta_A)\boldsymbol{v}_A + \eta_A\boldsymbol{u}_X$$

where $\eta_A > 0$ is a small constant. The corresponding perturbation in the matrix reads

$$\begin{aligned}
\boldsymbol{A}_{\eta_A}^c - \boldsymbol{A}^c &= \lambda_A(\tilde{\boldsymbol{v}}_A\tilde{\boldsymbol{v}}_A^T - \boldsymbol{v}_A\boldsymbol{v}_A^T) \\
&= \lambda_A\eta_A\left(\boldsymbol{v}_A\boldsymbol{u}_X^T + \boldsymbol{u}_X\boldsymbol{v}_A^T\right) + \mathcal{O}\left(\eta_A^2\right).
\end{aligned}$$

The first-order perturbation of the leading eigenvector yields

$$\boldsymbol{v}_1(\boldsymbol{A}_{\eta_A}^c) - \boldsymbol{v}_1(\boldsymbol{A}^c) = \lambda_A\eta_A\sum_{j\neq 1}\frac{\boldsymbol{v}_j(\boldsymbol{A}^c)^T\left(\boldsymbol{v}_A\boldsymbol{u}_X^T + \boldsymbol{u}_X\boldsymbol{v}_A^T\right)\boldsymbol{v}_A}{\lambda_A - \lambda_j}\boldsymbol{v}_j(\boldsymbol{A}^c) + \mathcal{O}\left(\eta_A^2\right).$$

Since $\boldsymbol{v}_j(\boldsymbol{A}^c)^T\boldsymbol{v}_A = 0$ for $j > 1$, we have

$$\begin{aligned}
&\left\langle \boldsymbol{v}_1(\boldsymbol{A}_{\eta_A}^c) - \boldsymbol{v}_1(\boldsymbol{A}^c), \tilde{\boldsymbol{y}}\right\rangle \\
&= \lambda_A\eta_A\sum_{j\neq 1}\frac{\tilde{\boldsymbol{y}}^T\boldsymbol{v}_j(\boldsymbol{A}^c)\boldsymbol{v}_j(\boldsymbol{A}^c)^T\boldsymbol{u}_X\boldsymbol{v}_A^T\boldsymbol{v}_A}{\lambda_A - \lambda_j} + \mathcal{O}\left(\eta_A^2\right) \\
&= \lambda_A\eta_A c_1\sum_{j\neq 1}\frac{\left(\tilde{\boldsymbol{y}}^T\boldsymbol{v}_j(\boldsymbol{A}^c)\right)^2}{\lambda_A - \lambda_j} + \lambda_A\eta_A c_2\sum_{j\neq 1}\frac{\tilde{\boldsymbol{y}}^T\boldsymbol{v}_j(\boldsymbol{A}^c)\boldsymbol{v}_j(\boldsymbol{A}^c)^T\boldsymbol{q}_X}{\lambda_A - \lambda_j} + \mathcal{O}\left(\eta_A^2\right) \quad (6)
\end{aligned}$$

where $c_1 = \sqrt{1 - \frac{\gamma(1+\mu)}{\mu(\mu+\gamma)}}$ and $c_2 = \sqrt{\frac{\gamma(1+\mu)}{\mu(\mu+\gamma)}}$. From the BBP transition we know that when $\lambda > 1$ we have $\langle \tilde{\boldsymbol{y}}, \boldsymbol{v}_A\rangle^2 = 1 - \frac{1}{\lambda^2} + \mathcal{O}_p(1)$. Consequently, it follows that $\sum_{j\neq 1}\left(\tilde{\boldsymbol{y}}^T\boldsymbol{v}_j(\boldsymbol{A}^c)\right)^2 = \frac{1}{\lambda^2} + \mathcal{O}_p(1)$. Since the edge of the bulk of spiked matrices still follows the Tracy–Widom distribution (Benaych-Georges et al., 2011), i.e., $\lambda_2 = 2 + \mathcal{O}_p\left(N^{-\frac{2}{3}}\right)$ and $\lambda_N = -2 + \mathcal{O}_p\left(N^{-\frac{2}{3}}\right)$, we have

$$\frac{1}{\frac{1}{\lambda} + \lambda + 2} + \mathcal{O}_p\left(N^{-\frac{2}{3}}\right) < \frac{1}{\lambda_A - \lambda_j} < \frac{1}{\frac{1}{\lambda} + \lambda - 2} + \mathcal{O}_p\left(N^{-\frac{2}{3}}\right) \quad \text{for} \quad j > 1.$$

Therefore the first term in (6) can be bounded as $\sum_{j\neq 1}\frac{\left(\tilde{\boldsymbol{y}}^T\boldsymbol{v}_j(\boldsymbol{A}^c)\right)^2}{\lambda_A - \lambda_j} \overset{a.s}{>} \frac{1}{\lambda_A^2}\frac{1}{\frac{1}{\lambda}+\lambda+2}$ when $N \to \infty$. For the second term, we note that the vector $\boldsymbol{q}_X$ is independent of $\boldsymbol{z} := \sum_{j\neq 1}\frac{\tilde{\boldsymbol{y}}^T\boldsymbol{v}_j(\boldsymbol{A}^c)}{\lambda_A - \lambda_j}\boldsymbol{v}_j(\boldsymbol{A}^c)$. Each eigenvector $\boldsymbol{v}_j(\boldsymbol{A}^c)$ is uniformly distributed on $\mathbb{S}_{\boldsymbol{v}_A^{\perp}}^{N-2}$ and $\{\boldsymbol{v}_j(\boldsymbol{A}^c)\}_{j=2}^N$ is an orthogonal basis of $\mathbb{R}_{\boldsymbol{v}_A^{\perp}}^{N-1} = \{\boldsymbol{v} : \boldsymbol{v} \in \mathbb{R}^N \mid \boldsymbol{v}^T\boldsymbol{v}_A = 0\}$. Therefore, for large $N$, $\tilde{\boldsymbol{y}}^T\boldsymbol{v}_j(\boldsymbol{A}^c)$ is approximately independent of each element in $\boldsymbol{v}_j(\boldsymbol{A}^c)$. More precisely, the entries of $\boldsymbol{z}$ are of the order of $\mathcal{O}_p\left(\frac{1}{\sqrt{N}}\right)$, and thus $\sum_{j\neq 1}\frac{\tilde{\boldsymbol{y}}^T\boldsymbol{v}_j(\boldsymbol{A}^c)\boldsymbol{v}_j(\boldsymbol{A}^c)^T\boldsymbol{q}_X}{\lambda_A - \lambda_j} = \langle\boldsymbol{q}_X, \boldsymbol{z}\rangle = \mathcal{O}_p\left(\frac{1}{\sqrt{N}}\right)$. Summarizing, we get

$$\left\langle\boldsymbol{v}_1(\boldsymbol{A}_{\eta_A}^c) - \boldsymbol{v}_1(\boldsymbol{A}^c), \tilde{\boldsymbol{y}}\right\rangle \overset{a.s.}{>} \frac{\sqrt{1 - \frac{\gamma(1+\mu)}{\mu(\mu+\gamma)}}}{(\lambda + 1/\lambda)(\lambda + 1/\lambda + 2)}\eta_A + \mathcal{O}\left(\eta_A^2\right) \quad \text{when} \quad N \to \infty.$$

A similar strategy can be used to show that $\langle\boldsymbol{u}_1(\boldsymbol{X}_{\eta_X}), \tilde{\boldsymbol{y}}\rangle \overset{a.s}{>} \langle\boldsymbol{u}_1(\boldsymbol{X}), \tilde{\boldsymbol{y}}\rangle$. $\qquad\square$

Table 6: Properties of the real-world benchmark datasets. For directed graphs we transform the graph to undirected in all experiments. $\mathcal{H}(\mathcal{G})$ indicates the homophily measure.

| Dataset | Classes | Features | Nodes | Edges | Directed | $\mathcal{H}(\mathcal{G})$ |
|---|---|---|---|---|---|---|
| Cora | 7 | $1,433$ | $2,708$ | $5,278$ | False | 0.810 |
| Citeseer | 6 | $3,703$ | $3,327$ | $4,552$ | False | 0.736 |
| PubMed | 3 | 500 | $19,717$ | $44,324$ | False | 0.802 |
| Computers | 10 | 767 | $13,752$ | $245,861$ | False | 0.777 |
| Photo | 8 | 745 | $7,650$ | $119,081$ | False | 0.827 |
| Chameleon | 6 | $2,325$ | $2,277$ | $31,371$ | True | 0.231 |
| Squirrel | 5 | $2,089$ | $5,201$ | $198,353$ | True | 0.222 |
| Actor | 5 | 932 | $7,600$ | $26,659$ | True | 0.219 |
| Texas | 5 | $1,703$ | 183 | 279 | True | 0.087 |
| Cornell | 5 | $1,703$ | 183 | 277 | True | 0.127 |
| Roman-empire | 18 | 300 | $22,662$ | $32,927$ | False | 0.047 |
| Amazon-ratings | 5 | 300 | $24,492$ | $93,050$ | False | 0.380 |
| Minesweeper | 2 | 7 | $10,000$ | $39,402$ | False | 0.683 |
| Tolokers | 2 | 10 | $11,758$ | $519,000$ | False | 0.595 |
| Questions | 2 | 301 | $48,921$ | $153,540$ | False | 0.840 |
| Penn94 | 2 | $4,814$ | $41,554$ | $1,362,229$ | False | 0.470 |
| Twitch-gamers | 2 | 7 | $168,114$ | $6,797,557$ | False | 0.545 |

## A.4 DATASETS

Table 6 shows the properties of the real-world datasets used. We also provide the homophily measure $\mathcal{H}(\mathcal{G})$ proposed in Pei et al. (2020), which we compute using the build-in function of Pytorch Geometric (Fey and Lenssen, 2019). For the cSBM, following (Chien et al., 2021), we choose $N = 5000$, $F = 2000$ and thus have $\gamma = \frac{N}{F} = 2.5$. Since the threshold to recover communities in cSBM is $\lambda^2 + \mu^2/\gamma > 1$ (Deshpande et al., 2018), we use a margin such that $\lambda^2 + \mu^2/\gamma = 1 + \epsilon$. We choose the same $\epsilon = 3.25$ as Chien et al. (2021) in all our experiments to be above the detection threshold and $d = 5$ to obtain a sparse graph to be close to the properties of real-world graphs. From the recovery threshold, we can parameterize the resulting arc of an ellipse with $\lambda \geq 0$ and $\mu \geq 0$ using $\phi = \arctan(\lambda\sqrt{\gamma}/\mu)$. Table 7 shows the parameters $\mu^2$ and $\lambda$ and the homophily measure $\mathcal{H}(\mathcal{G})$ for the different values of $\phi$.

## A.5 ADDITIONAL RESULTS

We provide a number of additional experiments which did not fit in the main text. These include more experiments on additional heterophilic datasets from Platonov et al. (2023), results for the homophilic datasets in the dense splitting, more experiments with DIGL (Gasteiger et al., 2019), more alignment results and results for synthetic and real-world data using spectral clustering with and w/o JDR. The clustering experiments in particular allow an interpretation of how JDR works: Applying it to a graph increases its "spectral clusterability".

### A.5.1 ADDITIONAL HETEROPHILIC DATASETS

In order to get a more comprehensive picture of the performance of JDR, we also test JDR on all the datasets proposed there by Platonov et al. (2023). Table 8 shows the results on these datasets using their original splits and comparing DIGL, FoSR, BORF and JDR. In general, the performance increases are relatively small for all methods, but overall JDR still performs best. It achieves significant performance increases on Tolokers and Questions. For Minesweeper none of the methods is really able to improve performance. The reason for this is in the synthetic design of its graph and the features: The graph does not contain any information about the labels, as it only connects neighboring cells (it is solely a computational graph). The same is partially true for node features which indeed contain information about neighboring mines, but only for $50\%$ of the nodes. This renders JDR unsuitable, which is also reflected in an interesting way in the experiments: We found

Table 7: Properties of the synthetic datasets generated from the cSBM with $\epsilon = 3.25$. $\mathcal{H}(\mathcal{G})$ indicates the homophily measure.

| $\phi$ | $\mu^2$ | $\lambda$ | $\mathcal{H}(\mathcal{G})$ |
|---|---|---|---|
| $-1.0$ | 0.0 | $-2.06$ | 0.039 |
| $-0.875$ | 0.40 | $-2.02$ | 0.049 |
| $-0.75$ | 1.56 | $-1.90$ | 0.076 |
| $-0.625$ | 3.28 | $-1.71$ | 0.119 |
| $-0.5$ | 5.31 | $-1.46$ | 0.170 |
| $-0.375$ | 7.35 | $-1.15$ | 0.241 |
| $-0.25$ | 9.07 | $-0.79$ | 0.325 |
| $-0.125$ | 10.22 | $-0.40$ | 0.408 |
| 0.0 | 10.63 | 0.0 | 0.496 |
| 0.125 | 10.22 | 0.40 | 0.583 |
| 0.25 | 9.07 | 0.79 | 0.671 |
| 0.375 | 7.35 | 1.15 | 0.751 |
| 0.5 | 5.31 | 1.46 | 0.837 |
| 0.625 | 3.28 | 1.71 | 0.879 |
| 0.75 | 1.56 | 1.90 | 0.925 |
| 0.875 | 0.40 | 2.02 | 0.955 |
| 1.0 | 0.0 | 2.06 | 0.963 |

Table 8: Comparison of DIGL, FoSR, BORF and JDR on real-world heterophilic datasets from Platonov et al. (2023): Mean accuracy (%) and ROC AUC for imbalanced Minesweeper, Tolokers and Questions$\pm$ 95% confidence interval. Best average accuracy in **bold**. OOM indicates an out-of-memory error.

| Method | Roman-empire | Amazon-ratings | Minesweeper | Tolokers | Questions |
|---|---|---|---|---|---|
| GCN | 78.64±0.42 | 46.19±0.58 | **90.08±0.31** | 84.61±0.59 | 75.31±0.81 |
| GCN+DIGL | 75.32±0.61 | 45.92±0.41 | 88.16±0.57 | 81.62±0.59 | 73.35±0.64 |
| GCN+FoSR | 78.58±0.43 | 46.30±0.44 | 90.07±0.51 | 84.50±0.47 | 75.51±0.73 |
| GCN+BORF | 78.66±0.42 | 46.44±0.54 | 90.06±0.38 | OOM | OOM |
| GCN+JDR | **78.86±0.48** | **46.47±0.67** | 90.01±0.32 | **84.73±0.45** | **77.52±0.63** |
| GPRGNN | 71.46±0.29 | 45.84±0.21 | 87.80±0.51 | 72.01±0.65 | 65.30±1.01 |
| GPRGNN+DIGL | 71.59±0.37 | **46.43±0.34** | **87.96±0.50** | 73.09±1.16 | 69.98±0.49 |
| GPRGNN+FoSR | 71.44±0.30 | 45.94±0.36 | 87.83±0.58 | 72.72±0.69 | 65.45±0.68 |
| GPRGNN+BORF | 71.46±0.26 | 45.79±0.33 | 87.81±0.51 | OOM | OOM |
| GPRGNN+JDR | **71.85±0.31** | 46.19±0.24 | 87.91±0.49 | **75.54±0.73** | **73.60±0.86** |

that the choice of hyperparameters has hardly any influence on performance. But also the results of the other rewiring methods indicate that they cannot be applied here. The graph is a standard grid-graph, which should not exhibit any interesting geometric properties and not contain any insights about the labels. Overall, this is a typical error case for any rewiring method. But it could also be discussed to what extent this dataset is an interesting *graph* dataset for node classification at all, since the connectivity does not contain any information about the labels.

### A.5.2 HOMOPHILIC DATASETS IN THE DENSE SPLITTING

Table 9 shows the results of DIGL, FoSR, BORF and JDR on real-world homophilic datasets in the *dense* splitting. The improvements of rewiring are smaller overall compared to the sparse splitting, but all four methods are able to improve it in most cases. With GCN as the downstream model, DIGL now performs best. JDR can still achieve the best result on two out of five data sets. When using GPRGNN as downstream model, JDR performs best on three out of five datasets. DIGL and FoSR are still able to achieve small performance improvements on most datasets and both rank first place on one dataset. BORF, on the other hand, is not able to improve the performance in most cases. This suggests that a more powerful GNN architecture benefits less from DIGL, FoSR or BORF, while

Table 9: Comparison of DIGL, BORF and JDR on real-world homophilic datasets using the *dense* splitting: Mean accuracy (%) $\pm$ 95% confidence interval. Best average accuracy in **bold**.

| Method | Cora | CiteSeer | PubMed | Computers | Photo |
|---|---|---|---|---|---|
| GCN | 88.14±0.27 | 79.02±0.25 | 86.14±0.10 | 89.03±0.12 | 94.07±0.10 |
| GCN+DIGL | 88.74±0.28 | 79.13±0.27 | **87.81±0.09** | **90.34±0.12** | **94.87±0.10** |
| GCN+FoSR | 88.09±0.28 | 79.23±0.25 | 86.14±0.10 | 88.98±0.12 | 94.04±0.09 |
| GCN+BORF | 88.18±0.24 | 79.17±0.24 | 86.14±0.10 | 89.14±0.11 | 94.00±0.10 |
| GCN+JDR | **88.76±0.25** | **80.25±0.27** | 86.20±0.10 | 88.93±0.13 | 94.20±0.08 |
| GPRGNN | 88.57±0.0.25 | 79.42±0.30 | 89.16±0.15 | 88.95±0.18 | 94.49±0.11 |
| GPRGNN+DIGL | 88.49±0.24 | 79.62±0.29 | 88.89±0.16 | **90.15±0.14** | 94.27±0.10 |
| GPRGNN+FoSR | 88.37±0.25 | 79.75±0.31 | **89.28±0.17** | 88.85±0.19 | 94.50±0.10 |
| GPRGNN+BORF | 88.56±0.27 | 79.39±0.31 | 89.04±0.18 | 88.90±0.19 | 94.52±0.10 |
| GPRGNN+JDR | **89.33±0.25** | **81.00±0.28** | 89.24±0.15 | 87.35±0.32 | **94.78±0.08** |

JDR can still improve it even further. The computer dataset is an exception for both downstream GNNs, JDR is not really able to improve the performance at all, while DIGL can clearly improve it.

### A.5.3   MLP with JDR

We design JDR with the aim to denoise the (possibly) complementary information in graph and features. This is based on the claim that a GNN is the method of choice when both the graph and the features contain valuable information, as it can utilize both. The experiments on cSBM in the main text show the ability of JDR to transfer information between the two. This becomes visible especially in the corner cases, were either the graph or the features do not contain any information about the labels. In this case the only way to improve is by transferring label information from one source to the other. To investigate this further, we test an MLP with JDR and compare the results with only the MLP and the GNNs. Since JDR can transfer the information from the graph to the features, an MLP should be able to perform similar to the GNNs. Therefore, we tune the hyperparameters of JDR with the MLP downstream model on the synthetic cSBM datasets and the real-world datasets from the main text.

The results on the cSBM data are shown in Figure 6. They show that combining an MLP with JDR clearly outperforms GCN in the heterophilic regime and performs very similar to GPRGNN. If the GNNs are also combined with JDR, they generally again provide superior performance compared to MLP+JDR. In the weak graph regime they have a huge performance advantage of about 20%, since JDR cannot improve the feature quality for the MLP in these cases. In the heterophilic regime, MLP+JDR is still comparable to GCN+JDR, which is not the case for the homophilic regime. A similar behavior (without JDR) has been observed in the literature before, e.g. by Ma et al. (2022) and is related to the limited ability of GCN to deal with heterophilic graphs. GPRGNN, however, does not show this limitation and provides superior performance across all datasets. Notably, all these findings do directly translate to the real-world datasets. The results on the real-world data can be found in Table 10 in the homophilic case and in Table 11 for the heterophilic case. For the homophilic datasets, MLP+JDR shows clear performance increases but cannot beat any of the GNN or GNN+JDR baselines. On the heterophilic graph, the MLP already outperforms GCN on three out of five datasets and with JDR on all datasets. GCN+JDR regains superior performance on three of the five datasets, but only GPRGNN+JDR outperforms the MLP+JDR on all datasets. Overall, this supports the claim that a GNN like GPRGNN is the method of choice when both the graph and the features contain valuable information.

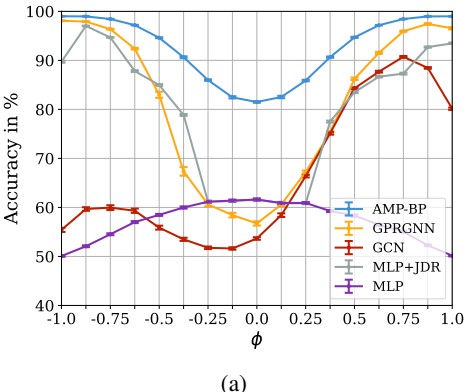 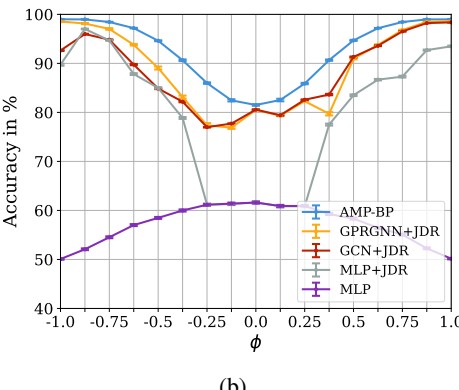

|     |     |
|:---:|:---:|
| (a) | (b) |

Figure 6: Comparison of MLP and the GNNs on the cSBM datasets in the sparse splitting. Comparisons of MLP and MLP+JDR with the GNNs (a) and GNN+JDR (b). The error bars indicate the 95% confidence interval. The MLP outperforms the GNNs in the very weak graph regime. Combining it with JDR clearly beats GCN (especially in the heterophilic regime) and performs very similar to GPRGNN. If the GNNs are combined with JDR, they generally again provide superior performance compared to MLP+JDR.

Table 10: Results of MLP and JDR on real-world homophilic dataset using the sparse splitting: Mean accuracy (%) $\pm$ 95% confidence interval. Best average accuracy in **bold**.

| Method | Cora | CiteSeer | PubMed | Computers | Photo |
|---|---|---|---|---|---|
| MLP | 50.79±0.73 | 50.29±0.48 | 79.73±0.13 | 73.17±0.31 | 80.88±0.33 |
| MLP+JDR | 62.66±0.61 | 61.55±0.32 | 80.86±0.12 | 80.65±0.24 | 88.34±0.45 |
| GCN | 77.26±0.35 | 67.16±0.37 | 84.22±0.09 | 84.42±0.31 | 91.33±0.29 |
| GCN+JDR | 79.96±0.26 | **69.35±0.28** | 84.79±0.08 | 85.66±0.36 | 92.52±0.23 |
| GPRGNN | 79.65±0.33 | 67.50±0.35 | 84.33±0.10 | 84.06±0.48 | 92.01±0.41 |
| GPRGNN+JDR | **80.77±0.29** | 69.17±0.30 | **85.05±0.08** | 84.77±0.35 | **92.68±0.25** |

Table 11: Results of MLP and JDR on real-world heterophilic dataset using the dense splitting: Mean accuracy (%) $\pm$ 95% confidence interval. Best average accuracy in **bold**.

| Method | Chameleon | Squirrel | Actor | Texas | Cornell |
|---|---|---|---|---|---|
| MLP | 49.07±0.57 | 28.19±0.40 | 38.54±0.30 | 91.16±0.79 | 88.19±0.74 |
| MLP+JDR | 70.48±0.46 | 59.18±0.31 | 39.50±0.26 | 91.16±0.77 | 88.47±0.77 |
| GCN | 67.65±0.42 | 57.94±0.31 | 34.00±0.31 | 75.62±1.12 | 64.68±1.25 |
| GCN+JDR | 69.76±0.50 | **61.76±0.39** | 40.47±0.31 | 85.12±0.74 | 84.51±1.06 |
| GPRGNN | 69.15±0.51 | 53.44±0.37 | 39.52±0.22 | 92.82±0.67 | 87.79±0.89 |
| GPRGNN+JDR | **71.00±0.50** | 60.62±0.38 | **41.89±0.24** | **93.85±0.54** | **89.45±0.84** |

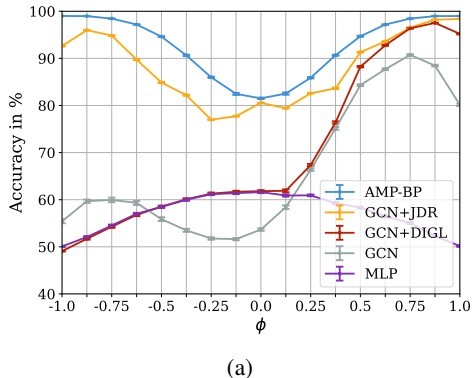 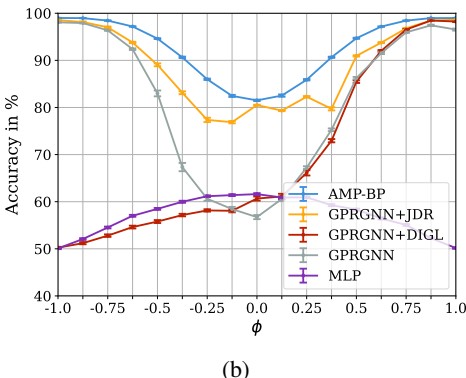

(a)                                    (b)

Figure 7: Comparison of DIGL (Gasteiger et al., 2019) and JDR on the cSBM datasets in the sparse splitting. Results for (a) GCN and (b) GPRGNN as downstream models. The error bars indicate the $95\%$ confidence interval. As expected, DIGL is not really able to improve the performance of the GNNs in the heterophilic regime. It achieves the greatest improvement in the weak-graph regime and for strongly homophilic graphs, especially using GCN as downstream model. Another interesting observation is that for GCN and $\phi < 0.25$ the curve of MLP corresponds exactly to the one of GCN+DIGL. The reason for this is that the hyperparameters found for DIGL ensure that the graph is ignored ($\alpha = 1.0$), which means that the GCN then collapses to a simple MLP. For the more powerfull GPRGNN, on the other hand, DIGL is generally hardly able to improve performance, while JDR clearly increases the performance across all $\phi$.

### A.5.4 COMBINING JDR AND DIGL

We compare our method to DIGL (Gasteiger et al., 2019) in the main text. We use the personalized PageRank (PPR) diffusion kernel and the same top-$64$ values sparsening method as in JDR in all experiments. Figure 7 shows the additional results for DIGL on the synthetic datasets from the cSBM. Table 12 shows the results on the real-world homophilic datasets in the sparse splitting and Table 13 on the heterophilic datasets in the dense splitting. Here, in addition to the individual results for JDR and DIGL, the results for a combination of the two methods are also shown. For this purpose, the graph was first denoised with JDR and then diffused with DIGL. To do this, we fixed the hyperparameters of JDR and then tuned the parameter $\alpha$ of DIGL. We think this is interesting as both methods enhance the graph in different ways and thus should be combinable. In principle, this should also be possible for a combination of JDR with BORF or FoSR, but so far we have not been able to get this to work.

**Homophilic datasets.** For the homophilic datasets, both DIGL and JDR can improve the results when GCN is used as a downstream model. Still, DIGL is outperformed by JDR on four out the five datasets. The two methods can be combined on three of the five data sets to achieve even better results. This gives empirical support for the assumption that the two methods use a distinct way of performing rewiring in this case and a combination therefore can further increase accuracy. The picture is somewhat different for GPRGNN as a downstream model. The improvements for DIGL are significantly smaller here, whereas JDR shows clear improvements across all datasets. This suggests that a more powerful GNN architecture benefits less from DIGL, while JDR can still improve it even further. A combination of the two methods does not lead to an increase in performance here. Although the performance is still significantly better compared to no rewiring or just DIGL, JDR alone usually performs better.

**Heterophilic datasets.** Since DIGL rewires the graph by adding edges between nodes with short diffusion distance, it is expected to perform poorly on the heterophilic datasets. The results using GCN show that this is only true for Chameleon and Squirrel, while for Actor, Texas and Cornell there are still considerable improvements. For the datasets Texas and Cornell, DIGL even achieve the best results. JDR, on the other hand, improves performance across datasets and GNNs. This is also in line with the finding on the cSBM in Figure 7a. However, we can also see that DIGL can not really improve performance of GPRGNN. JDR, on the other hand, can still achieve an improvement

Table 12: Comparison of DIGL and JDR on real-world homophilic dataset using the sparse splitting: Mean accuracy (%) $\pm$ 95% confidence interval. Best average accuracy in **bold**.

| Method | Cora | CiteSeer | PubMed | Computers | Photo |
|---|---|---|---|---|---|
| GCN | 77.26±0.35 | 67.16±0.37 | 84.22±0.09 | 84.42±0.31 | 91.33±0.29 |
| GCN+DIGL | 79.27±0.26 | 68.03±0.33 | 84.60±0.09 | **86.00±0.24** | 92.00±0.23 |
| GCN+JDR | 79.96±0.26 | **69.35±0.28** | 84.79±0.08 | 85.66±0.36 | 92.52±0.23 |
| GCN+JDR+DIGL | **80.48±0.26** | 69.19±0.29 | **84.83±0.10** | 84.78±0.34 | **92.69±0.22** |
| GPRGNN | 79.65±0.33 | 67.50±0.35 | 84.33±0.10 | 84.06±0.48 | 92.01±0.41 |
| GPRGNN+DIGL | 79.77±0.30 | 67.50±0.35 | 84.72±0.10 | **86.25±0.28** | 92.31±0.25 |
| GPRGNN+JDR | **80.77±0.29** | 69.17±0.30 | **85.05±0.08** | 84.77±0.35 | **92.68±0.25** |
| GPRGNN+JDR+DIGL | 80.55±0.27 | **69.47±0.27** | 84.87±0.10 | 85.98±0.21 | 92.67±0.27 |

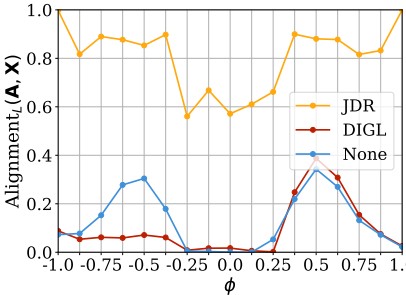

Figure 8: Alignment of the leading eigenspaces according to (3) for graphs from the cSBM with different $\phi$. We compare JDR to DIGL and no rewiring.

across all datasets. A combination of DIGL and JDR is generally not particularly useful in this scenario, likely because DIGL has difficulties on the heterophilic datasets anyway.

Table 13: Comparison of DIGL and JDR on real-world heterophilic dataset using the dense splitting: Mean accuracy (%) $\pm$ 95% confidence interval. Best average accuracy in **bold**.

| Method | Chameleon | Squirrel | Actor | Texas | Cornell |
|---|---|---|---|---|---|
| GCN | 67.65±0.42 | 57.94±0.31 | 34.00±0.31 | 75.62±1.12 | 64.68±1.25 |
| GCN+DIGL | 58.04±0.48 | 39.64±0.34 | 39.57±0.29 | **91.05±0.73** | **88.49±0.74** |
| GCN+JDR | **69.76±0.50** | **61.76±0.39** | 40.47±0.31 | 85.12±0.74 | 84.51±1.06 |
| GCN+JDR+DIGL | 66.06±0.43 | 36.62±0.29 | 40.30±0.27 | 88.90±0.73 | 88.06±0.77 |
| GPRGNN | 69.15±0.51 | 53.44±0.37 | 39.52±0.22 | 92.82±0.67 | 87.79±0.89 |
| GPRGNN+DIGL | 66.57±0.46 | 42.98±0.37 | 39.61±0.21 | 91.11±0.72 | 88.06±0.81 |
| GPRGNN+JDR | **71.00±0.50** | **60.62±0.38** | **41.89±0.24** | **93.85±0.54** | **89.45±0.84** |
| GPRGNN+JDR+DIGL | 70.07±0.44 | 59.37±0.35 | 41.57±0.20 | 91.52±0.70 | 87.77±1.81 |

### A.5.5 ALIGNMENT

Here, we give a more detailed view on how much JDR actually increases alignment on cSBM and real-world datasets compared to the baseline methods. For cSBM, we can see in Figure 8 that DIGL only increases alignment in the homophilic regime. In the heterophilic regime it clearly decreases alignment. We expect this because it promotes connections among nodes at short diffusion distance. Also the random teleport probability found on these datasets is 1.0, which results in a random uniformly connected graph. Similar to the results on the real-world datasets Cornell and Texas from the main text, we can see this in the classification performance in Figure 7. In the heterophilic regime, the performance of DIGL matches exactly the MLP, while in the homophilic regime, we can see some performance increases. With GPRGNN, DIGL is not really able to improve performance at all (except for $\phi = 0$). In Figure 9, we can see that JDR increases alignment across all settings and more strongly than the baseline methods on the real-world graphs (except for Citeseer).

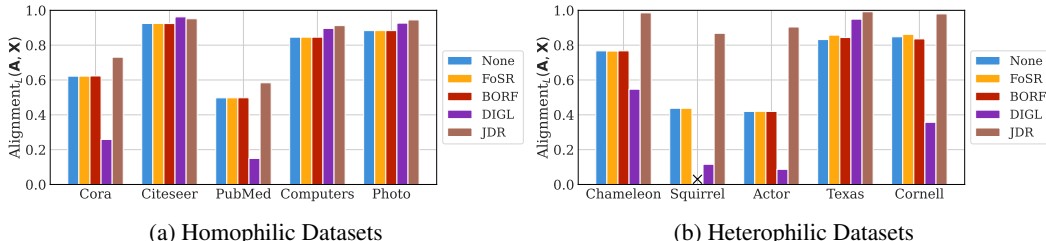

|                              |                                |
| :--------------------------: | :----------------------------: |
| (a) Homophilic Datasets      | (b) Heterophilic Datasets      |

Figure 9: Alignment of the leading eigenspaces of graphs from homophilic (a) and heterophilic (b) real-world datasets. We compare the original graph (None) to the output of DIGL, BORF and JDR with the hyperparameters found on GCN. JDR increases the alignment in all settings and achieves the maximum alignment among rewiring algorithms in all settings except on the Citeseer dataset.

DIGL also increases alignment on many homophilic graphs and on heteophilic Texas. We would like to note that when only rewiring a graph, increasing the alignment might not always be the best thing to do: If the graph is very good, it would be the features that should be made more aligned to the graph. But only JDR is able to do this, as it also denoises the features as well. It is also interesting that DIGL decreases the alignment for Cora and PubMed, but still achieves a good performance. This indicates that DIGL in this case improves the graph in a different way than JDR. So here, it should be possible to combine both methods to achieve even better performance. And indeed, A.5.4 shows that this is exactly the case for Cora, PubMed and Photo on GCN. FoSR and BORF do not visibly change the alignment, since their modifications to the graph are usually too small.

### A.5.6 SPECTRAL CLUSTERING

In addition to the GNNs as a downstream algorithm, we also experiment with spectral clustering (SC). Spectral clustering either works with an existing graph, or a k-nearest neighbor graph is created from given (high-dimensional) node features. Then the $k$ largest eigenvectors of the graph are calculated (the first one is usually omitted as it is a constant vector) and their entries are then used as coordinates for a k-means clustering of the nodes into $k$ classes. We show that JDR using the hyperparameters found with GCN as a downstream model, improves the performance of a spectral clustering algorithm acting directly on $A$ or $X$. This indicates a close connection between GCNs and spectral clustering such that a good denoised graph for GCN is also a good graph for spectral clustering. Intuitively, since spectral clustering is related to the graph cut, this means that in this case the classes are connected with fewer edges, making them easier to cluster based on the cut.

Table 14: Results on real homophilic datasets using spectral clustering: Mean accuracy (%) and best result in **bold**. Here, all methods use the hyperparameters found using GCN as downstream algorithm.

| Method       | Cora      | CiteSeer  | Pubmed    | Computers | Photo     |
| :----------- | :-------- | :-------- | :-------- | :-------- | :-------- |
| SC(A)        | 33.83     | 24.16     | 58.94     | 37.35     | 30.58     |
| SC(A)+DIGL   | 29.54     | 22.18     | 59.65     | 61.55     | 25.41     |
| SC(A)+FoSR   | 33.83     | 24.61     | 58.72     | 36.97     | 30.54     |
| SC(A)+BORF   | 35.01     | 25.22     | 58.90     | 37.35     | 33.37     |
| SC(A)+JDR    | **67.76** | **63.36** | **72.90** | **62.29** | **65.67** |
| SC(X)        | 29.76     | 45.57     | 60.45     | 28.53     | 48.46     |
| SC(X)+JDR    | **34.68** | **45.90** | **60.47** | **28.55** | **48.58** |

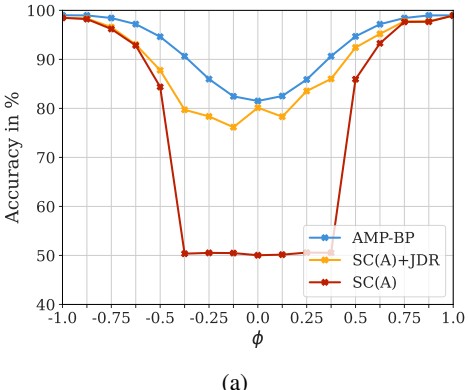 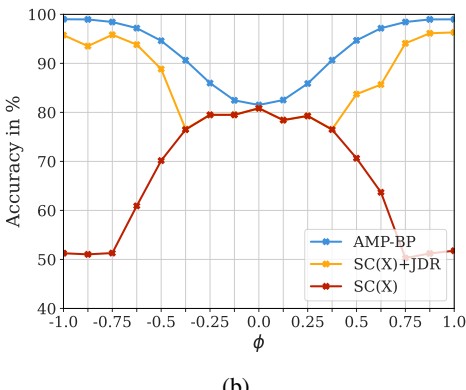

(a)  (b)

Figure 10: Separate results for using spectral clustering on a rewriting only $A$ (a) and denoising only $X$ (b) compared to full JDR . Note that for $\phi \in \{0.5, 0.625, 0.875\}$ we had to use additional graphs generated using cSBM with an average node degree of $d = 10$ for spectral clustering of $A$ to work in general and for $\phi = 0.875$ also for JDR. The reason for this is that the graph is very sparse so it is not necessarily connected such that there is no guarantee that spectral clustering works. However a larger node degree does not improve the performance of spectral clustering in general, while it may for GNNs.

Table 15: Results on real heterophilic datasets using spectral clustering: Mean accuracy (%) and best result in **bold**. Here, all methods use the hyperparameters found using GCN as downstream algorithm.

|  | Chameleon | Squirrel | Actor | Texas | Cornell |
|---|---|---|---|---|---|
| SC(A) | 31.71 | 22.40 | 25.92 | 48.09 | 39.89 |
| SC(A)+DIGL | **32.06** | 22.69 | 25.91 | 43.72 | 40.44 |
| SC(A)+FoSR | 31.44 | **24.46** | 25.93 | 55.19 | 38.80 |
| SC(A)+BORF | 31.97 | OOM | 25.97 | **56.83** | 43.17 |
| SC(A)+JDR | 31.36 | 22.15 | **28.63** | 52.46 | **44.26** |
| SC(X) | 23.54 | 20.17 | **31.01** | 49.18 | 45.36 |
| SC(X)+JDR | **24.59** | **21.03** | 23.99 | **55.74** | **49.73** |

**cSBM**. Figure 10 displays the results of spectral clustering with and w/o JDR. Figure 10a indicates the expected behavior that spectral clustering using $A$ performs particularly poorly in the weak graph regime, since in this case there is hardly any information about the labels in $A$. By using JDR, this limitation is completely removed and the performance is close to AMP-BP across all $\phi$. The rewired graph now contains more information about the labels, which was previously only available in $X$. For spectral clustering of $X$ in Figure 10b, the relation is exactly the other way around. In the strong heterophilic or homophilic regime the performance is poor since most information is contained in the graph structure. Using JDR this limitation is removed and the performance becomes closer to AMP-BP across all $\phi$. Although a slight denoising of $X$ by $A$ would be possible for $\phi = \pm 0.375$, there is no performance advantage here and these settings now show the weakest performance across all $\phi$.

**Real-world Datasets**. For the real world datasets, we compare the spectral clustering of $A$ using the different rewiring methods DIGL, FoSR , BORF and JDR. For the spectral clustering of $X$ we can only evaluate JDR. Again we use the hyperparameters found using GCN as downstream model. The results in Table 14 on homophilic datasets show a significant benefit of using JDR across all datasets. FoSR, BORF and DIGL are also able to improve the performance in some settings but not very consistently. There are also performance improvements across all datasets for the spectral clustering of $X$ with JDR, but these are significantly smaller. This indicates that the rewiring of the graph has a significantly greater influence on performance here than the denoising of the features. It also gives an indication of how JDR works on real-world data: It increases "spectral

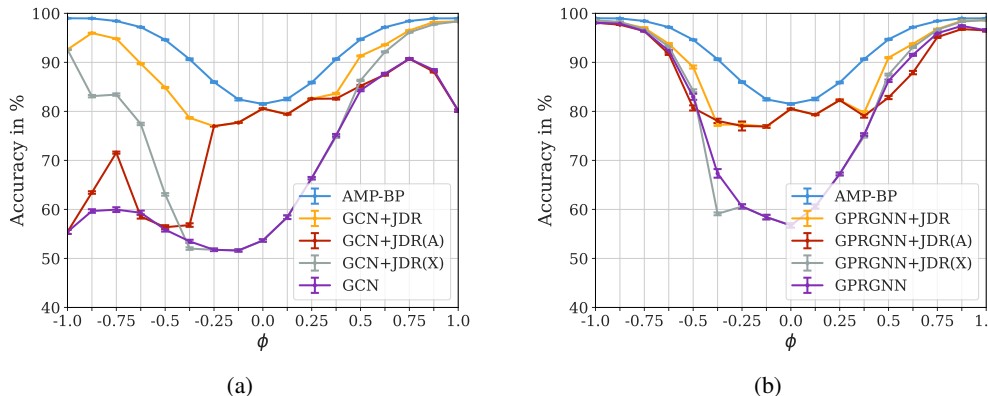

Figure 11: Separate results for rewiring only $A$ and denoising only $X$ compared to full JDR. Results for (a) GCN and (b) GPRGNN. The error bars indicate the $95\%$ confidence interval.

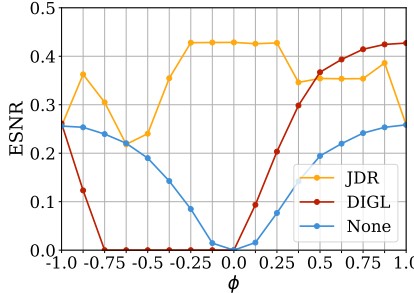

Figure 12: ESNR for graphs from the cSBM with different $\phi$. We compare JDR to DIGL and no rewiring.

clusterability" of the graph. Table 15 shows the results for the heterophlic datasets. The results here are much more inconsistent. It is striking that DIGL improves Chameleon and Squirrel, while it has actually worsened performance for GCN. BORF can improve the performance on Texas and Cornell by a large margin, although DIGL and JDR perform better with the GCN. FoSR improves the performance of Squirrel. For the results of JDR, it is worth looking at them together with the spectral clustering of $X$. On Chameleon and Squirrel the performance decreases for $A$ but clearly increases for $X$. On Texas and Cornell it is improved in all cases, but on $A$ not as strongly as for BORF. On Actor, the performance for $X$ has dropped, while JDR is also the only method that really improves the result for $A$. To summarize, the improvements for JDR can be assigned to one of the two sources of information, either $A$ or $X$, for each dataset.

### A.5.7 EVALUATING GRAPH DENOISING VIA ESNR

As described in the main text, the Graph Propensity Score (GPS) algorithm (Dong and Kluger, 2023), together with the edge signal-to-noise ratio (ESNR) metric introduced in the same paper, are related to our work. Similar to the GPS algorithm, which uses the ESNR to denoise the graph, JDR denoises the graph but also the features. These two strategies are different, but they are both based on the cSBM. The main drawback of GPS is that it does not consider the possibility to also denoise the features using the graph and it is further limited to low-SNR graphs and to GCN as downstream model. We are interested in the the ESNR metric as it quantifies the noise in the graph.

In Figure 12, Table 16 and Table 17 we compare the achieved ESNR values of JDR with the baselines and the original graphs. The results for cSBM in Figure 12 show that JDR is able to denoise the graph in all cases and DIGL in the strongly homophilic regime. However, comparing the ESNR curve to the actual GCN performance in Figure 7, there is no clear connection apart from the general trend. This was claimed to be more visible in the paper by Dong and Kluger (2023), especially for cSBM on which it is based on. The results on homophilic real-world datasets in Table 16 follow

Table 16: ESNR results of the original and rewired homophilic real-world graphs using JDR and the baseline rewiring methods. Largest value in **bold**.

| Method | Cora | CiteSeer | PubMed | Computers | Photo |
|---|---|---|---|---|---|
| None | 0.2964 | 0.1976 | 0.2701 | 0.6502 | 0.7145 |
| DIGL | **0.6888** | **0.5989** | **0.5490** | **0.7368** | **0.7407** |
| FoSR | 0.2965 | 0.1988 | 0.2700 | 0.6500 | 0.7145 |
| BORF | 0.2958 | 0.1948 | 0.2700 | 0.6502 | 0.7144 |
| JDR | 0.6160 | 0.5754 | 0.5291 | 0.7069 | 0.7155 |

Table 17: ESNR results of the original and rewired heterophilic real-world graphs using JDR and the baseline rewiring methods. Largest value in **bold**.

| Method | Chameleon | Squirrel | Actor | Texas | Cornell |
|---|---|---|---|---|---|
| None | 0.5199 | **0.4680** | 0.0546 | 0.0585 | 0.0388 |
| DIGL | 0.5082 | 0.3884 | 0.0244 | 0.1442 | 0.1195 |
| FoSR | **0.5199** | 0.4679 | 0.0546 | 0.0620 | 0.0389 |
| BORF | 0.5199 | - | 0.0545 | 0.0593 | 0.0423 |
| JDR | 0.4473 | 0.2156 | **0.1703** | **0.2729** | **0.2828** |

this trend. Indeed both JDR and DIGL are able to decrease the noise in the graph, but DIGL does so best on all five datasets, while the GCN results show a benefit of JDR over DIGL on four out of five datasets (see Table 2). For the heterophilic real-world datasets the results are even more inconsistent. JDR is able to decrease the graph noise on three datasets, but two of them not being the ones where GCN performs best (see Table 3). Also on Chameleon on Squirrel, no method is able to improve the ESNR. We suspect that this behavior occurs due to the role of features which is not captured by the ESNR. Checking the results in the paper by Dong and Kluger (2023), there are cases where the ESNR is not sensitive, e.g. on the Chameleon dataset, which suggests that in such cases, denoising the features is more beneficial than focusing only on the graph structure. In fact, this is in line with our findings in the ablations in Figure 13 and Figure 14, where the denoising of the features improves more than the denoising of the graph on the Chameleon dataset. Overall, the ESNR is generally able to quantify the denoising of the graphs in most cases, but a more direct connection to GCN performance requires further research.

Finally, there are significant differences between the GPS approach by Dong and Kluger (2023) and our work, most importantly in that we consider both graph and feature information. Moreover, JDR improves performance of GNNs on real-world graph datasets where GPS does not provide any improvements (homophilic datasets). Ablations on these datasets (again Figure 13 and Figure 14) indicate that JDR still achieves the most improvement by denoising the graph rather than the features in a way that does not seem to be captured by the ESNR. We think that this is because ESNR does not consider the SNR of the features. In the experiments of Dong and Kluger (2023) with edge dropout on real-world data, the performance barely decreases for datasets like PubMed. The reason for this is that the features already contain a lot of information and making the graph noisier via edge dropout does not spoil this. This is also related to the discussion of GNN vs. MLP (see A.5.3), where an MLP on PubMed already performs reasonably well (see e.g. Table 10.

## A.6   ABLATIONS

We perform several ablations of our method to investigate what happens in different scenarios and what effects changes in parameters have. First, we present our ablations of the JDR method. We show separate results for denoising only the graph JDR(A) or the features JDR(X) using the GNNs on the cSBM and real-world data. Also, we show several ablations of the hyperparameters of JDR. We therefore use a dataset created from cSBM, the homophilic dataset Cora and the heterophilic dataset Chameleon. Ablations on the real-world datasets are performed for all hyperparameters of JDR and show its robustness to change in these parameters.

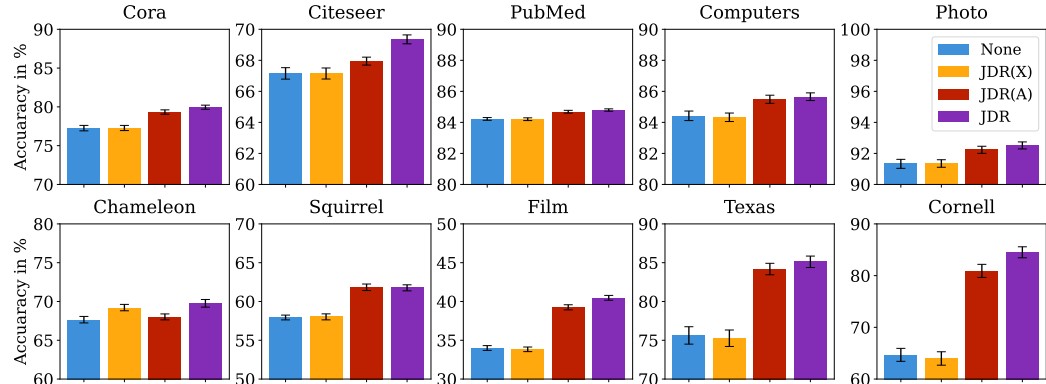

Figure 13: Average accuracy of GCN on all real-world datasets tested for denoising only the features JDR(X), rewiring only the graph JDR(A) and joint denoising and rewiring JDR.

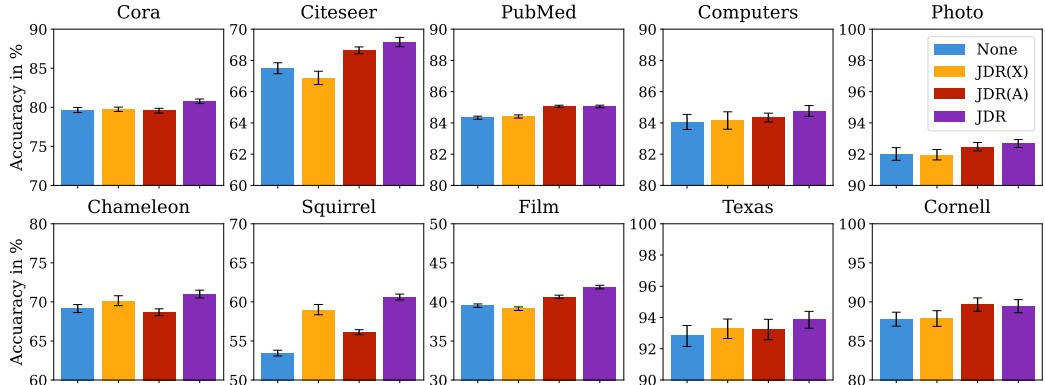

Figure 14: Average accuracy of GPRGNN on all real-world datasets tested for denoising only the features JDR(X), rewiring only the graph JDR(A) and joint denoising and rewiring JDR. It can be observed that for most datasets, the major improvement is achieved by JDR(A). Only for Squirrel and Chameleon it is JDR(X). In most cases using JDR on both $X$ and $A$ achieves the best performance.

**JDR.** The main motivation for these ablations is to show how much impact the denoising of $A$ and $X$ respectively have on the results for a dataset and how big the additional benefit is to do this jointly. Therefore we look at the results if we denoise only the graph JDR(A) or the features JDR(X). Doing this for the cSBM in Figure 11, we can observe the expected behavior, which is particularly pronounced for GCN in Figure 11a. In the weak graph regime, the performance increase results purely from denoising $A$, so JDR(A) achieves the same performance as JDR. The same holds for JDR(X) in the strong homophilic regime and for $\phi = -1.0$. In the remaining intermediate regimes, we can often observe a performance benefit of both JDR(A) and JDR(X), which becomes much stronger when we combine both. This benefit of combining both is particularly pronounced for $\phi = -0.375$, where JDR(X) alone even reduces performance, while JDR clearly improves performance. In Figure 11b, we can basically observe the same behavior, but less strongly pronounced. Moreover, it happens in several cases here, again especially in the intermediate regime, that the performance is reduced by JDR(X), but improved for the joint denoising.

Figure 13 and Figure 14 show the same investigation for the real-world datasets using GCN and GPRGNN, respectively. In most cases, the greater performance gain results from JDR(A) and the joint denoising performs best. Only for the datasets Chameleon for both GNNs and Squirrel for GPRGNN, the denoising of $X$ has the greater influence. Also the case where the denoising of $X$ reduces the performance, but a joint denoising performs best, occurs here, e.g. for Citeseer or

Table 18: Comparison for JDR using hyperparameters tuned on different downstream models. The "*" indicates that the hyperparameters of JDR where tuned using the same GNN as downstream model, no symbol mean that the respective other GNN model was used. Results on real-world homophilic datasets using sparse splitting (2.5%/2.5%/95%): Mean accuracy (%) $\pm$ 95% confidence interval. Best average accuracy in **bold**.

| Method | Cora | CiteSeer | PubMed | Computers | Photo | ↑Gain |
|---|---|---|---|---|---|---|
| GCN | 77.26$\pm$0.35 | 67.16$\pm$0.37 | 84.22$\pm$0.09 | 84.42$\pm$0.31 | 91.33$\pm$0.29 | - |
| GCN+JDR* | **79.96**$\pm$**0.26** | **69.35**$\pm$**0.28** | **84.79**$\pm$**0.08** | **85.66**$\pm$**0.36** | **92.52**$\pm$**0.23** | 1.59 |
| GCN+JDR | 78.85$\pm$0.29 | 69.11$\pm$0.28 | 84.20$\pm$0.09 | 85.61$\pm$0.21 | 92.25$\pm$0.25 | 1.13 |
| GPRGNN | 79.65$\pm$0.33 | 67.50$\pm$0.35 | 84.33$\pm$0.10 | 84.06$\pm$0.48 | 92.01$\pm$0.41 | - |
| GPRGNN+JDR | 80.47$\pm$0.33 | 68.94$\pm$0.29 | **85.17**$\pm$**0.09** | 84.64$\pm$0.25 | 92.64$\pm$0.21 | 0.86 |
| GPRGNN+JDR* | **80.77** $\pm$**0.29** | **69.17**$\pm$**0.30** | 85.05$\pm$0.08 | **84.77**$\pm$**0.35** | **92.68**$\pm$**0.25** | 0.98 |

Cornell. Overall, this confirms that our method indeed performs *joint* denoising, especially when both graph and node contain relevant information both benefit from denoising.

**Hyperparameters.** In Table 18 we show how the downstream GNN performs if JDR was tuned on a different downstream GNN. We use GCN and GPRGNN for this. The results show that the hyperparameters of JDR are quite robust to different GNN downstream models as it achieves similar gains using the respective other hyperparameters. Another way to show the robustness of JDR is to perform ablations of the actual hyperparameters. To do this, we first look at a data set generated from the cSBM and examine the influence of the number of denoising iterations $K$ and the number of entries of the adjacency matrix to be retained $A_k$. Figure 15 show the results of this study. As expected increase both results in better performance but will also increase the computational complexity. Based on this, we choose $A_k = 64$ for all experiments as a good trade-off between computational and memory cost and accuracy over different numbers of denoising iterations. We also investigate this effect together with the rest of the hyperparameters for the real-world datasets Cora in Figure 16 and Chameleon in Figure 17. We again examine the number of denoising iterations $K$ and the number of entries of the adjacency matrix to be retained $A_k$. Additionally, we study the interpolation ratios $\eta_X$ and $\eta_A$ and the number of eigenvectors for the denoising $L_X$ and $L_A$. Both are analyzed relative to the value found by random search and for both $A$ and $X$ at the same time. For the interpolation ratios $\eta_X$ and $\eta_A$, we show the influence of using only a reduced number of digits of the best found value (0 corresponds to no denoising) and for the number of eigenvectors $L_X$ and $L_A$ we test different offsets (0 corresponding to the best value found using random search). Overall, we con observe a strong robustness to changes in the hyperparameters. Only the number of denoising iterations $K$ should not be too high for the heterophilic data set Chameleon.

## A.7 HYPERPARAMETERS

In this section we list all the hyperparameters used for the experiments to ensure the reproducibility of the results. They are also included in the code. In all experiments we use the Adam optimizer and the standard early stopping after 200 epochs from (Chien et al., 2021). Whenever we use a GCN, it uses two layers, a hidden dimension of 64 and dropout with 0.5. Whenever we use GPRGNN, we use a polynomial filter of order 10 (corresponding to 10 hops) and a hidden dimension of 64. For JDR, we always keep the 64 largest entries of the rewired adjacency matrix $\tilde{A}$ per node. We justify this choice by the ablation in Figure 15.

**cSBM**. For synthetic data from the cSBM, we generally follow the hyperparameters from (Chien et al., 2021). GCN uses a learning rate of 0.01 and weight decay with $\lambda = 0.0005$. GPRGNN also uses a $\lambda = 0.0005$ and both use ReLU non-linearity. On homophilic graphs ($\phi \geq 0$), GPRGNN uses a learning rate of 0.01, a weight initialization $\alpha = 0.1$ and dropout with 0.5. For heterophilic graphs, it uses a learning rate of 0.05, $\alpha = 1.0$ and dropout 0.7. The hyperparameters for JDR on the cSBM are shown in Table 22. We only tuned them using GCN as a downstream model, so for GPRGNN+JDR we use the same ones.

**Real-world Datasets.** For the real-world datasets, the remaining hyperparameters for GCN are displayed in Table 19 and for GPRGNN in Table 21. The hyperparameters for JDR can be found

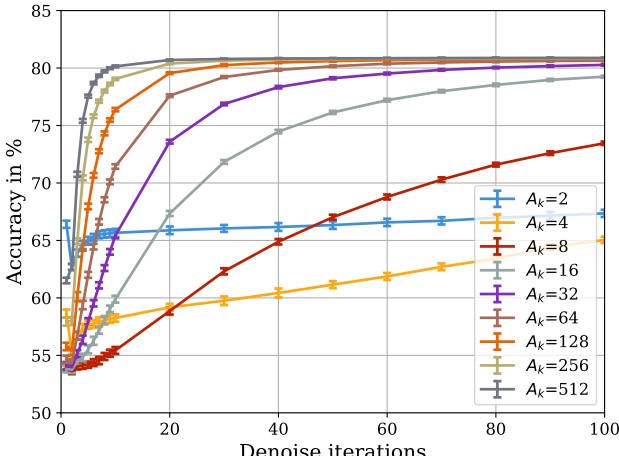

Figure 15: Average accuracy of GCN on cSBM with $\phi = 0.0$ for different numbers of denoise iterations and different numbers of entries $A_k$ to keep for each node in the rewired adjacency matrix. Error bars indicating the $95\%$ confidence interval over 100 runs.

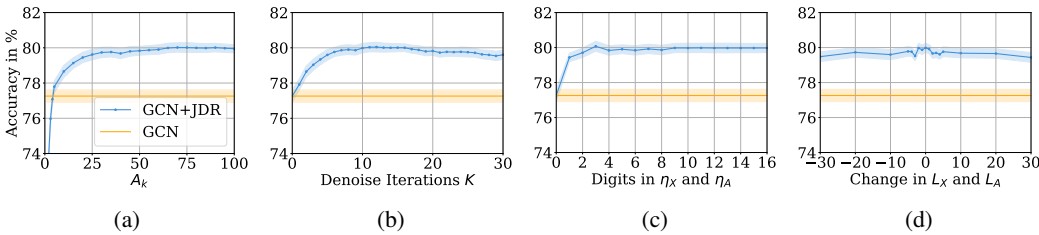

Figure 16: Ablations of GCN+JDR on the homophilic dataset Cora compared to the result for GCN. The light shaded ares indicate the $95\%$ confidence interval. Ablations on the number number of entries chosen per node for the adjacency $A_k$, the number of denoise iterations $K$, the number of interpolations digits for the $\eta$ values and the number of eigenvectors $L_\square$ used. All other parameters are kept constant. In all cases we can see that JDR is quite robust to changes in all of its hyperparameters.

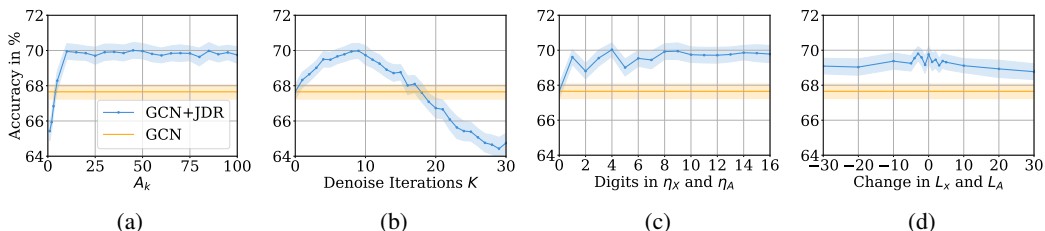

Figure 17: Ablations of GCN+JDR on the heterophilic dataset Chameleon compared to the result for GCN. The light shaded ares indicate the $95\%$ confidence interval. We perfrom the same ablations as for Cora. In all cases except the number of denoising iterations, we can see that JDR is quite robust to changes in all of its hyperparameters.

in Table 23 and Table 24. For the rewiring method BORF, we list its hyperparameters in Table 25 and Table 26. For DIGL, we always use the PPR kernel and sparsify the result by keeping the top-64 values for a weighted adjacency matrix. The values for the random-teleport probabililty $\alpha$ and number of iterations for FoSR are listed in Table 27 and Table 28.

Table 19: Hyperparameters of GCN. All models use 2 layers, a hidden dimension of 64 and dropout with 0.5. Different type of weight decay and early stopping from (Gasteiger et al., 2019) was used, if these provided a better performance then using the standard setting in Chien et al. (2021). The same holds for feature normalization, which was used by default in Chien et al. (2021) for GPRGNN.

| **Dataset** | Lr | Normalize $X$ | $\lambda_1$ | $\lambda_1$ layer | Early stopping |
|---|---|---|---|---|---|
| Cora | 0.01 | False | 0.05 | First | GPRGNN |
| Citeseer | 0.01 | True | 0.0005 | All | GPRGNN |
| PubMed | 0.01 | True | 0.0005 | All | GPRGNN |
| Computers | 0.01 | False | 0.0005 | All | GPRGNN |
| Photo | 0.01 | False | 0.0005 | All | GPRGNN |
| Chameleon | 0.05 | True | 0.0 | All | DIGL |
| Squirrel | 0.05 | True | 0.0 | All | DIGL |
| Actor | 0.01 | False | 0.0005 | All | DIGL |
| Texas | 0.05 | True | 0.0005 | All | GPRGNN |
| Cornell | 0.05 | True | 0.0005 | All | GPRGNN |

Table 20: Hyperparameters of GCN for the larger heterophilic datasets. All models use a hidden dimension of 64, batch norm, no weight decay and the early stopping from Chien et al. (2021).

| **Dataset** | Lr | # layers | Dropout | Residuals |
|---|---|---|---|---|
| Questions | 0.005 | 5 | 0.2 | True |
| Penn94 | 0.001 | 2 | 0.5 | False |
| Twitch-gamers | 0.01 | 4 | 0.5 | False |

Table 21: Hyperparameters of GPRGNN. All models use 10 hops and a hidden dimension of 64.

| **Dataset** | Lr | Normalize $X$ | $\alpha$ | $\lambda_1$ | Dropout | Early stopping |
|---|---|---|---|---|---|---|
| Cora | 0.01 | True | 0.1 | 0.0005 | 0.5 | GPRGNN |
| Citeseer | 0.01 | True | 0.1 | 0.0005 | 0.5 | GPRGNN |
| PubMed | 0.05 | True | 0.2 | 0.0005 | 0.5 | GPRGNN |
| Computers | 0.01 | False | 0.1 | 0.0005 | 0.5 | GPRGNN |
| Photo | 0.01 | False | 0.5 | 0.0 | 0.5 | GPRGNN |
| Chameleon | 0.05 | False | 1.0 | 0.0 | 0.7 | DIGL |
| Squirrel | 0.05 | True | 0.0 | 0.0 | 0.7 | GPRGNN |
| Actor | 0.01 | True | 0.9 | 0.0 | 0.5 | GPRGNN |
| Texas | 0.05 | True | 1.0 | 0.0005 | 0.5 | GPRGNN |
| Cornell | 0.05 | True | 0.9 | 0.0005 | 0.5 | GPRGNN |
| Questions | 0.005 | False | 1.0 | 0.0 | 0.2 | GPRGNN |
| Penn94 | 0.01 | False | 0.1 | 0.0001 | 0.2 | GPRGNN |
| Twitch-gamers | 0.001 | False | 0.5 | 0.0001 | 0.2 | GPRGNN |

Table 22: Hyperparameters for GCN on the cSBM in the sparse splitting. For all homophilic datasets the eigenvalues are ordered by value and for all heterophilic datasets they are ordered by absolute value. In all setting we keep the $64$ largest entries of the rewired adjacency matrix $\tilde{A}$ per node. Interpolation ratios $\eta$ are rounded to three digits from the best values found by the random search.

| $\phi$ | JDR | | | | | | DIGL |
|---|---|---|---|---|---|---|---|
| | $K$ | $L_A$ | $L_X$ | $\eta_A$ | $\eta_{X_1}$ | $\eta_{X_2}$ | $\alpha$ |
| $-1.0$ | 28 | – | 10 | – | 0.482 | 0.916 | 1.0 |
| $-0.875$ | 41 | 5 | 8 | 0.101 | 0.479 | 0.858 | 1.0 |
| $-0.75$ | 40 | 6 | 9 | 0.042 | 0.498 | 0.846 | 1.0 |
| $-0.625$ | 48 | 6 | 8 | 0.036 | 0.453 | 0.862 | 1.0 |
| $-0.5$ | 50 | 9 | 10 | 0.189 | 0.412 | 0.991 | 1.0 |
| $-0.375$ | 48 | 8 | 10 | 0.879 | 0.973 | 0.773 | 1.0 |
| $-0.25$ | 80 | 1 | 1 | 1.000 | – | – | 1.0 |
| $-0.125$ | 80 | 1 | 1 | 1.000 | – | – | 1.0 |
| $0.0$ | 80 | 1 | 1 | 1.000 | – | – | 0.95 |
| $0.125$ | 76 | 1 | – | 0.650 | – | – | 1.0 |
| $0.25$ | 33 | 1 | – | 0.951 | – | – | 0.5 |
| $0.375$ | 18 | 10 | 10 | 0.856 | 0.023 | 0.228 | 0.05 |
| $0.5$ | 18 | 10 | 9 | 0.415 | 0.263 | 0.880 | 0.05 |
| $0.625$ | 22 | 8 | 7 | 0.264 | 0.340 | 0.807 | 0.05 |
| $0.75$ | 15 | 7 | 9 | 0.056 | 0.474 | 0.778 | 0.05 |
| $0.875$ | 16 | 10 | 8 | 0.035 | 0.228 | 0.981 | 0.05 |
| $1.0$ | 80 | – | 1 | – | 1.000 | 1.000 | 0.05 |

Table 23: Hyperparameters of JDR for all real-world datasets in the dense splitting. Following the findings from cSBM for all homophilic datasets the eigenvalues are ordered by value and for all heterophilic datasets they are ordered by absolute value. In all setting we keep the $64$ largest entries of the rewired adjacency matrix $\tilde{A}$ per node. Interpolation ratios $\eta$ are rounded to three digits from the best values found by the random search.

| Dataset | GNN | | | | | | GPRGNN | | | | | |
|---|---|---|---|---|---|---|---|---|---|---|---|---|
| | $K$ | $L_A$ | $L_X$ | $\eta_A$ | $\eta_{X_1}$ | $\eta_{X_2}$ | $K$ | $L_A$ | $L_X$ | $\eta_A$ | $\eta_{X_1}$ | $\eta_{X_2}$ |
| Cora | 10 | 1853 | 38 | 0.066 | 0.173 | 0.071 | 10 | 772 | 76 | 0.027 | 0.434 | 0.005 |
| Citeseer | 15 | 578 | 1330 | 0.460 | 0.173 | 0.049 | 4 | 1390 | 1169 | 0.345 | 0.099 | 0.585 |
| PubMed | 12 | 8 | 53 | 0.316 | 0.004 | 0.187 | 1 | 1772 | 919 | 0.197 | 0.893 | 0.034 |
| Computers | 3 | 718 | 975 | 0.398 | 0.021 | 0.068 | 7 | 583 | 1533 | 0.468 | 0.062 | 0.127 |
| Photo | 6 | 467 | 1867 | 0.479 | 0.071 | 0.344 | 4 | 433 | 1719 | 0.413 | 0.115 | 0.231 |
| Chameleon | 7 | 41 | 1099 | 0.066 | 0.375 | 0.975 | 3 | 31 | 1331 | 0.063 | 0.486 | 0.755 |
| Squirrel | 2 | 4 | 1941 | 0.404 | 0.011 | 0.022 | 2 | 53 | 1210 | 0.234 | 0.495 | 0.964 |
| Actor | 29 | 896 | 14 | 0.298 | 0.235 | 0.219 | 11 | 1171 | 791 | 0.476 | 0.028 | 0.251 |
| Texas | 20 | 21 | 183 | 0.514 | 0.028 | 0.836 | 1 | 109 | 36 | 0.182 | 0.004 | 0.214 |
| Cornell | 17 | 10 | 125 | 0.794 | 0.298 | 0.113 | 1 | 39 | 67 | 0.482 | 0.424 | 0.068 |
| Questions | 8 | 248 | 284 | 0.218 | 0.199 | 0.841 | 2 | 89 | 2 | 0.974 | 0.106 | 0.311 |
| Penn94 | 20 | 60 | 71 | 0.445 | 0.005 | 0.902 | 5 | 172 | 1851 | 0.422 | 0.094 | 0.138 |
| Twitch-gamers | 5 | 7 | 5 | 0.235 | 0.286 | 0.806 | 5 | 2 | 2 | 0.165 | 0.329 | 0.003 |

Table 24: Hyperparameters of JDR for all the homophilic datasets in the sparse splitting. Following the findings from cSBM for all homophilic datasets the eigenvalues are ordered by value and for all heterophilic datasets they are ordered by absolute value. In all setting we keep the 64 largest entries of the rewired adjacency matrix $\tilde{A}$ per node. Interpolation ratios $\eta$ are rounded to three digits from the best values found by the random search.

| Dataset | GNN | | | | | | GPRGNN | | | | | |
|---|---|---|---|---|---|---|---|---|---|---|---|---|
| | $K$ | $L_A$ | $L_X$ | $\eta_A$ | $\eta_{X_1}$ | $\eta_{X_2}$ | K | $L_A$ | $L_X$ | $\eta_A$ | $\eta_{X_1}$ | $\eta_{X_2}$ |
| Cora | 10 | 1853 | 38 | 0.066 | 0.173 | 0.071 | 10 | 772 | 76 | 0.027 | 0.434 | 0.005 |
| Citeseer | 15 | 578 | 1330 | 0.460 | 0.173 | 0.049 | 4 | 1390 | 1169 | 0.345 | 0.099 | 0.585 |
| PubMed | 12 | 8 | 53 | 0.316 | 0.004 | 0.187 | 1 | 1772 | 919 | 0.197 | 0.893 | 0.034 |
| Computers | 3 | 718 | 975 | 0.398 | 0.021 | 0.068 | 7 | 583 | 1533 | 0.468 | 0.062 | 0.127 |
| Photo | 6 | 467 | 1867 | 0.479 | 0.071 | 0.344 | 4 | 433 | 1719 | 0.413 | 0.115 | 0.231 |

Table 25: Hyperparameters for BORF for all real-world datasets in the dense splitting. OOM indicates an out-of-memory error.

| Dataset | GNN | | | GPRGNN | | |
|---|---|---|---|---|---|---|
| | # iterations | # added | # removed | # iterations | # added | # removed |
| Cora | 2 | 30 | 10 | 1 | 10 | 40 |
| Citeseer | 3 | 30 | 40 | 3 | 10 | 50 |
| PubMed | 2 | 0 | 30 | 3 | 20 | 40 |
| Computers | 1 | 20 | 40 | 3 | 20 | 30 |
| Photo | 2 | 40 | 20 | 3 | 50 | 50 |
| Chameleon | 2 | 50 | 30 | 1 | 10 | 30 |
| Squirrel | OOM | | | OOM | | |
| Actor | 2 | 40 | 50 | 2 | 10 | 50 |
| Texas | 1 | 40 | 10 | 2 | 40 | 50 |
| Cornell | 1 | 20 | 50 | 1 | 20 | 50 |

Table 26: Hyperparameters for BORF for the homophilic real-world datasets in the sparse splitting.

| Dataset | GNN | | | GPRGNN | | |
|---|---|---|---|---|---|---|
| | # iterations | # added | # removed | # iterations | # added | # removed |
| Cora | 2 | 10 | 40 | 2 | 30 | 50 |
| Citeseer | 3 | 50 | 40 | 1 | 20 | 50 |
| PubMed | 2 | 0 | 30 | 3 | 20 | 40 |
| Computers | 1 | 20 | 40 | 3 | 20 | 30 |
| Photo | 3 | 0 | 50 | 3 | 10 | 20 |

Table 28: Values of the hyperparameter $\alpha$ of DIGL and the number of iterations (# iter) of FoSR for the homophilic real-world datasets in the sparse splitting.

| Dataset | $\alpha$ DIGL | | $\alpha$ DIGL+JDR | | #iter FoSR | |
|---|---|---|---|---|---|---|
| | GCN | GPRGNN | GCN | GPRGNN | GCN | GPRGNN |
| Cora | 0.10 | 0.30 | 0.10 | 0.30 | 5 | 75 |
| Citeseer | 0.30 | 0.45 | 0.20 | 0.45 | 500 | 600 |
| PubMed | 0.35 | 0.60 | 0.40 | 0.60 | 50 | 75 |
| Computers | 0.05 | 0.65 | 0.15 | 0.30 | 250 | 800 |
| Photo | 0.20 | 0.50 | 0.10 | 0.50 | 5 | 500 |

Table 27: Values of the hyperparameter $\alpha$ of DIGL and the number of iterations (# iter) of FoSR for the real-world datasets in the dense splitting.

| Dataset | $\alpha$ DIGL | | $\alpha$ DIGL+JDR | | #iter FoSR | |
|---|---|---|---|---|---|---|
| | GCN | GPRGNN | GCN | GPRGNN | GCN | GPRGNN |
| Cora | 0.25 | 0.60 | 0.20 | 0.60 | 150 | 5 |
| Citeseer | 0.60 | 0.50 | 0.25 | 0.25 | 1000 | 1000 |
| PubMed | 0.60 | 0.50 | 0.60 | 0.65 | 10 | 5 |
| Computers | 0.05 | 0.60 | 0.10 | 0.65 | 500 | 600 |
| Photo | 0.30 | 0.70 | 0.20 | 0.75 | 25 | 250 |
| Chameleon | 0.15 | 0.50 | 0.55 | 0.40 | 10 | 5 |
| Squirrel | 0.05 | 0.15 | 0.10 | 0.20 | 5 | 10 |
| Actor | 1.00 | 0.60 | 0.20 | 0.05 | 10 | 150 |
| Texas | 1.00 | 0.00 | 0.20 | 0.20 | 50 | 75 |
| Cornell | 1.00 | 1.00 | 0.95 | 0.00 | 25 | 5 |
| Questions | 0.05 | 1.0 | — | — | 700 | 700 |
| Penn94 | 0.2 | 0.1 | — | — | 200 | 150 |
| Twitch-gamers | 0.15 | 0.25 | — | — | 5 | 100 |

## A.8 HARDWARE SPECIFICATIONS

Experiments on cSBM, Cora, Citeseer and Photo were conducted on an internal cluster with Nvidia Tesla V100 GPUs with 32GB of VRAM. The experiments on the remaining datasets (PubMed, Computers, Chameleon, Squirrel, Actor, Cornell and Texas) were performed using Nvidia A100 GPUs with 40GB or 80GB of VRAM. The larger VRAM is only necessary for GNN+JDR on PubMed and the larger heterophilic datasets from Lim et al. (2021); Platonov et al. (2023), because they have larger numbers of nodes in the graph (and we choose the top-64 edges per node after rewiring). Note that this could be reduced by sacrificing only a little bit of performance as shown in A.6. One experiment of training and testing on 100 random splits typically takes about $5 \, \text{min}$. For the standard benchmark graphs, the longest experiments with GPRGNN+JDR and a different early stopping condition take about $40 \, \text{min}$. The experiments on the large Twitch-gamers dataset take around $60 \, \text{min}$ (for 10 splits), but similar to DIGL they require a lot of standard memory (around $500\text{GB}$) while performing the decompositions.

## A.9 INSIGHTS FROM RANDOM MATRIX THEORY FOR ONE-LAYER GCNS

Following the derivation from Shi et al. (2024), we show empirically how denoising can reduce the empirical risk for a one-layer GCN without non-linearity. When the number of nodes $N$ goes to infinity and the average node degree satisfies some assumptions, we can apply the Gaussian adjacency equivalence conjecture. This allows us to replace the binary adjacency in the cSBM with a spiked non-symmetric Gaussian random matrix without changing the training and test loss in the

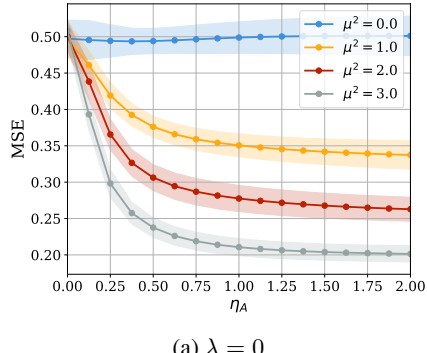 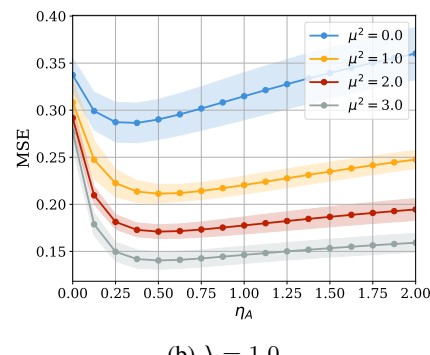

(a) $\lambda = 0$             (b) $\lambda = 1.0$

Figure 18: Experimental results on a non-symmetric gaussian cSBM with $N = 1000$ and $\gamma = 2$ with denoising of $\boldsymbol{A}$. We plot the MSE for different $\mu^2$ and $\lambda = 0.0$ in 18a and $\lambda = 1.0$ in 18b. Each data point is averaged over 10 independent trials and the standard deviation is indicated by the light shaded area.

limit. The equivalent adjacency reads

$$\boldsymbol{A} = \frac{\lambda}{N} \boldsymbol{y} \boldsymbol{y}^T + \boldsymbol{\Xi}_{gn} \tag{7}$$

where with $\boldsymbol{\Xi}_{gn}$ has i.i.d. centered normal entries with variance $1/N$. Similarly, we build the features matrix as

$$\boldsymbol{X} = \frac{\mu}{N} \boldsymbol{y} \boldsymbol{u}^T + \boldsymbol{\Xi}_x. \tag{8}$$

Compared to the standard cSBM formulation we rescale the variables $\sqrt{\mu\gamma} \to \mu$ and $\sqrt{F}\boldsymbol{u} \to \boldsymbol{u}$. Additionally, we define $\alpha = 1/\gamma = F/N$ and for simplicity, we consider the case $\boldsymbol{I}_{train} = \boldsymbol{I}$. The mean squared error (MSE) loss reads

$$L(\boldsymbol{\omega}) = \frac{1}{N} \|\boldsymbol{A}\boldsymbol{X}\boldsymbol{\omega} - \boldsymbol{y}\|_F^2 + \frac{r}{N} \|\boldsymbol{\omega}\|^2, \tag{9}$$

where $r$ is the parameter for the ridge part, $\boldsymbol{\omega}$ are the weights of the GCN and $\|\|_F$ indicates the Frobenius norm. For $N \to \infty$, the MSE concentrates, which means it is only a function of $\mu$, $\lambda$ and $\alpha$. For denoising $\boldsymbol{A}$ we do

$$\boldsymbol{A}_{\text{den}} = \boldsymbol{A} + \eta_A \boldsymbol{X} \boldsymbol{X}^T. \tag{10}$$

The idea is that although this leads to more noise terms, the signal strength of $\boldsymbol{y}\boldsymbol{y}^T$ is increased more. Instead of a weighting of $\frac{\lambda}{N}\boldsymbol{y}\boldsymbol{y}^T$, we now have $(\frac{\lambda}{N} + \eta_A \frac{\mu^2 F}{N})\boldsymbol{y}\boldsymbol{y}^T$. The new MSE also concentrates on a value determined by $\eta_A$. So, numerically, as shown in Figure 18, for any $\mu, |\lambda| > 0$ we can always find values of $\eta_A$ such that the MSE is decreased. For denoising $\boldsymbol{X}$ we do

$$\boldsymbol{X}_{\text{den}} = \boldsymbol{X} + \eta_X \boldsymbol{A} \boldsymbol{X} \tag{11}$$

and show in Figure 19 with the same argumentation as for $\boldsymbol{A}$ that an $\eta_X$ exists so that the MSE is reduced. Proof of both cases has yet to be provided and will be the subject of future work.

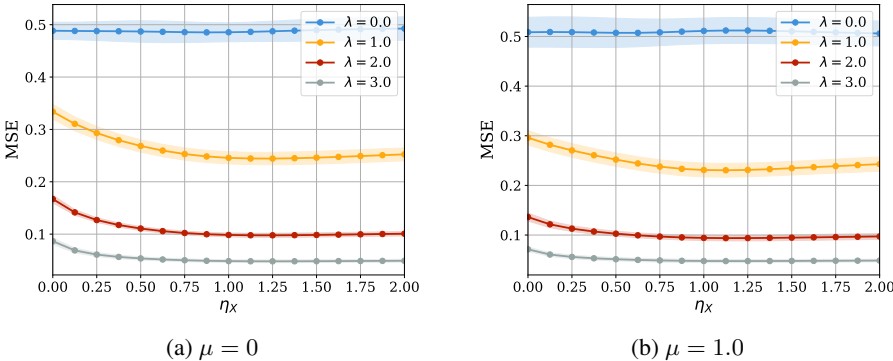

(a) $\mu = 0$ (b) $\mu = 1.0$

Figure 19: Experimental results on a non-symmetric gaussian cSBM with $N = 1000$ and $\gamma = 2$ with denoising of $\boldsymbol{X}$. We plot the MSE for different $\lambda$ and $\mu = 0.0$ in 19a and $\mu = 1.0$ in 19b. Each data point is averaged over 10 independent trials and the standard deviation is indicated by the light shaded area.

