# OpenReview forum: "Joint Graph Rewiring and Feature Denoising via Spectral Resonance"
_ICLR.cc/2025/Conference — ICLR 2025 Oral_

### Official Review · Reviewer_Fauu · 2024-10-26

**Soundness:** 3
**Presentation:** 2
**Contribution:** 2
**Rating:** 8
**Confidence:** 3

**Summary:**

The work proposes a method (JDR) to jointly perform graph rewiring and feature denoising. The method is derived from cSBM to maximize alignment between the eigenspaces of features and the graph. The authors tested the performance of the methods in various datasets and showed the advantages of the proposed JDR method.

**Strengths:**

1. The method is well inspired and can achieve better performance than several other graph rewiring methods.
2. An impressive amount of experiments were implemented.

**Weaknesses:**

1. The paper should better discuss previous works on GNN versus MLP, and the connection between heterophily and graph noise. [1-4] investigated the phenomena that MLP sometimes perform better than GNN (especially GCN) for heterophilic graphs. In particular, [4] proposed a metric that well correlates with empirical GNN performance, and also discussed the connection between heterophily and graph noise. A better discussion (acknowledgement?) of [4] is needed due to its high relevance to this paper.

2. The comparison between JDR and graph rewiring methods seems not perfectly fair as the authors also mention themselves. Moreover, if JDR indeed achieves optimal denoising, then the graph may no longer be needed in later training. This (JDR(X) + MLP) seems uncovered by the ablation settings.

3. The authors should check if the rewired graph structures degenerate, which may be a natural consequence of combining matrix factorization scheme and thresholding.

4. The hyperparameter tuning was not performed for the GNN backbone. This is questionable as the optimal backbone hyperparameter is anticipated to vary across different rewired graphs (and potentially different denoised features). This may matter because the performance gap between JDR and other methods seems small in most cases.

5. Some paragraphs are unclear and not readable. In particular, it is unclear what “findings” in the sentence “The Gaussian adjacency equivalence conjecture (Shi et al., 2024) suggests a generalization of the findings to the true binary adjacency case.” is referring to. The whole section (as well as the proof) needs to be polished.

[1] Ma, Yao, et al. "Is Homophily a Necessity for Graph Neural Networks?." International Conference on Learning Representations.

[2] Gomes, Diana, et al. "When Are Graph Neural Networks Better Than Structure-Agnostic Methods?." I Can't Believe It's Not Better Workshop: Understanding Deep Learning Through Empirical Falsification. 2022.

[3] Luan, Sitao, et al. "Revisiting heterophily for graph neural networks." Advances in neural information processing systems35 (2022): 1362-1375.

[4] Dong, Mingze, and Yuval Kluger. "Towards understanding and reducing graph structural noise for GNNs." International Conference on Machine Learning. PMLR, 2023.

**Questions:**

See above.

---

> ### Author Response · Authors · 2024-11-20
> **Reply to Reviewer Fauu (Part 1/2)**
>
> We thank the reviewer for their constructive comments and detailed feedback, particularly for drawing our attention to the topic of GNNs vs MLPs. In the following, we want to address the individual concerns mentioned by the reviewer. We will upload a revised paper soon.
>
> **Weaknesses:**
>
> > 1. The paper should better discuss previous works on GNN versus MLP, and the connection between heterophily and graph noise. [1-4] investigated the phenomena that MLP sometimes perform better than GNN (especially GCN) for heterophilic graphs. In particular, [4] proposed a metric that well correlates with empirical GNN performance, and also discussed the connection between heterophily and graph noise. A better discussion (acknowledgement?) of [4] is needed due to its high relevance to this paper.
>
> Thank you for bringing these works to our attention. We actually already cite the work of Dong and Kluger (2023) in our related work section, but we will highlight it more in the revised version in the context of graph denoising.
>
> As you suggest, MLPs indeed outperform GNNs in the weak graph regime; we have acknowledged that in the main text (line 354 ff.). But we emphasize that even in that regime, after applying JDR, GNN outperforms the MLP (cf. the discussion below on W.2). This suggests that in real-world datasets the graphs really do contain information that is complementary to features, and JDR can help tease it out. It means that GNNs can still perform strongly in this situation, provided that we first use this complementarity to denoise the graph.
>
>
> > 2. The comparison between JDR and graph rewiring methods seems not perfectly fair as the authors also mention themselves.
>
> You are right. We do state this since we aim to disclose the limitations of our method as transparently as possible.
>
> At the same time, since JDR is the first method to jointly denoise the graph and the features, there are no more direct comparisons that we can do. This is why we very explicitly state this under limitations. That is why many of our experiments focus on showing what advantage features bring to rewiring.
>
> We acknowledge that our wording may not reflect this clearly and will articulate it accordingly.
> The comparison is fair insofar as there is no restriction for other methods to use feature information---it's only that JDR is the only method that does. The real limitation of JDR is that it cannot be applied without node features. We will emphasize this in the revised manuscript.
>
> > Moreover, if JDR indeed achieves optimal denoising, then the graph may no longer be needed in later training.
>
> We do not think that JDR achieves optimal denoising (that does not even seem possible for real graphs). We will make this explicit in the revision. We do claim that it improves alignment between leading spectral spaces and consequently with the labels (as per Proposition 1 and numerical experiments). But even if denoising was in some sense optimal, due to the different inductive biases of different deep net architectures, it may still be favorable to provide different views of label information (as a graph and as features), since even optimal denoising cannot result in zero noise.
>
> As mentioned before, we consider cases where the graph and features contain complementary information. In this case, even an optimal, though graph-agnostic denoising, cannot restore the full information about the labels in the features (and vice versa). So we assume that by "optimal denoising" you refer to the case, where we also transfer information from graph to features (and vice versa).
> JDR is indeed designed such that it is able to perform this transfer due to it alternating joint denoising scheme. The experiments on cSBM verify this in the corner cases were either the graph or the features do not contain any information about the labels. In this case, the only way to improve is by transferring label information from one source to the other. However, this does not mean that the graph does not help for node classification anymore, which is visible for cSBM where GNN+JDR clearly outperforms the MLP in the very weak graph regime.

---

> ### Author Response · Authors · 2024-11-20
> **Reply to Reviewer Fauu (Part 2/2)**
>
> > This (JDR(X) + MLP) seems uncovered by the ablation settings.
>
> You are right. We ran new experiments corresponding to this ablation. Based on your W.2 we also became interested in how well an MLP with JDR can perform. The MLP+JDR experiments show to which extend JDR can transfer information from graph to features. These new results on the cSBM and real-world datasets can be found in Appendix A.5.3. We think they provide really interesting insight (so thanks again for suggesting them!), which we briefly present here. As you pointed out in the question and as we have seen in the main text, an MLP outperforms the GNNs in the weak graph regime. Now combining the MLP with JDR clearly outperforms GCN (especially in the heterophilic regime) and performs very similar to GPRGNN. However, if GPRGNN is combined with JDR, it again provides superior performance compared to MLP+JDR, **even in the heterophilic regime** which you highlighted. This supports our claim that GNNs can be the method of choice when the graph and the features contain valuable information, since they do use both. The described behaviors can be observed both on synthetic data from the cSBM and real-world datasets.
>
> > 3. The authors should check if the rewired graph structures degenerate, which may be a natural consequence of combining matrix factorization scheme and thresholding.
>
> Could you please elaborate what you mean by degenerate here? If you refer to the cases where the rewired graph splits into disconnected components or the corresponding matrices become low rank, we did not observe this with JDR. We do not threshold the spectrum (rather interpolate), but we do (hard) threshold the synthesized interpolated adjacency matrices: we retain the top-$k$ entries per node, resulting in the new, rewired edges.
>
> We should mention that we did observe one degenerate case on the Cornell and Texas graphs when we treated $k$ as a hyperparameter. The best choice for $k$ using a GCN as a downstream model ended up being close to $0$. This, however, did not happen with GPRGNN. This is compatible with our findings about the relative merits of GNNs and MLPs (also of W.1 and W.2) and with what we observed for DIGL on these datasets (l.426 ff.). Since these graphs are very noisy, removing edges is a good approach if the downstream algorithm is too weak to handle the high noise, which is the case for GCN. With better downstream algorithms which can handle the high noise, such as GPRGNN, the edge removal strategy is not optimal. We hope this speaks to your question, otherwise we'd be happy to discuss further.
>
> > 4. The hyperparameter tuning was not performed for the GNN backbone. This is questionable as the optimal backbone hyperparameter is anticipated to vary across different rewired graphs (and potentially different denoised features). This may matter because the performance gap between JDR and other methods seems small in most cases.
>
> We agree with the reviewer that we could achieve better results by also tuning the hyperparameters of the GNNs for each rewired graph. One could even go one step further and iteratively tune the parameters of JDR and the downstream GNN multiple times. We did not do it because it would **massively** increase the required GPU time and associated energy but it would not qualitatively change the findings. We agree with you that the results could only improve. Since we are not tuning the hyperparameters of the GNNs for all rewiring methods, we still achieve a fair comparison across methods.
>
> > 5. Some paragraphs are unclear and not readable. In particular, it is unclear what “findings” in the sentence “The Gaussian adjacency equivalence conjecture (Shi et al., 2024) suggests a generalization of the findings to the true binary adjacency case.” is referring to. The whole section (as well as the proof) needs to be polished.
>
> Thank you for pointing this out, we will rework these sections for clarity in the revision. The statement you quote in your question refers to an analytic technique used in the study of the cSBM (Deshpande et al., 2018). Random matrix theory gives us strong guarantees for Gaussian matrices, e.g., for Gaussian Orthogonal Ensembles (GOEs), but there are much fewer results for the actual binary adjacency matrices. The Gaussian adjacency equivalence conjecture (Shi et al., 2024) states that one can analyze a graph using a Gaussian adjacency matrix and then this analysis generalizes to the true binary adjacency. We use this reasoning to show that JDR improves alignment under the cSBM. We hope that this make our statement clearer; please do let us know if that is not the case.

---

> > ### Comment · Reviewer_Fauu · 2024-11-21
> >
> > Thanks for the rebuttal. I am happy with the response and appreciate the effort of authors in improving the work. In particular, the additional experiments of JDR + MLP indeed provides interesting and deep insights. I am raising my score to 8. The only remaining thing is, I was actually referring to the first part of Dong and Kluger (2023), that proposes a measure of graph noise and connects it to real graphs and GNN performance, instead of the latter common part about graph rewiring. It would be great if its connections to the paper is better discussed / clarified somehow.

---

> > > ### Author Response · Authors · 2024-12-02
> > > **Reply to Reviewer Fauu**
> > >
> > > Thank you for still more constructive feedback!
> > >
> > > We studied more carefully the first part of Dong and Kluger and agree that the ESNR metric is interesting and related to our work as it quantifies the noise in the graph. Similarly to their GPS algorithm which uses the ESNR to denoise the graph, JDR denoises the graph but also the features. The two strategies are different, but they are indeed both based on / inspired by the cSBM. We’ll make this clear in the revision.
> > >
> > > Checking the results on real-world datasets, there are cases where the ESNR is not sensitive, e.g. on the Chameleon dataset, which suggests that in such cases, denoising the features is more beneficial than focusing only on the graph structure. In fact, this is in line with our findings in the ablations in Figure 12 and 13, where the denoising of the features improves more than the denoising of the graph on the Chameleon dataset. We will further investigate these connections by using the ESNR metric to quantify how much denoising happens in the graph. We will add these results in the camera-ready version.
> > >
> > > Finally, there are significant differences between the work by Dong and Kluger and our work, most importantly in that we consider both graph and feature information. Moreover, JDR improves performance of GNNs on real-world graph datasets where GPS does not provide any improvements (homophilic datasets). Ablations on these datasets (again Figure 12 and 13) indicate that JDR still achieves the most improvement by denoising the graph rather than the features in a way that does not seem to be captured by the ESNR. We think that this is because ESNR does not consider the SNR of the features. In the experiments with edge dropout on real-world data, the performance barely decreases for datasets like PubMed. The reason for this is that the features already contain a lot of information and making the graph *noisier* via edge dropout does not spoil this. This is also related to the discussion of GNN vs. MLP, where an MLP on PubMed already performs reasonably well.

---

> > > > ### Comment · Reviewer_Fauu · 2024-12-02
> > > >
> > > > I'm satisfied with the response and very glad to see authors' effort even after increasing the scores. Adding the ESNR comparison will indeed further clarify the role of reducing graph noise in JDR, and some claims in the last paragraph of the response. I am already convinced that this paper presents an important improvement and different from existing works.

---

### Official Review · Reviewer_Pd13 · 2024-10-31

**Soundness:** 3
**Presentation:** 3
**Contribution:** 3
**Rating:** 8
**Confidence:** 3

**Summary:**

This work presents a structural rewiring algorithm, with consideration of structural alignment with node features, supported by theoretical investigation on cSBM model.

**Strengths:**

1.The overall structure is well organized and easy to understand, especially the diagram demonstrations are very helpful.
2.The problem formulation is clearly presented.
3.Experiments are comprehensive and convincing.

**Weaknesses:**

1. The theoretical framework seems convincing to me, while how the real data sets fit the parametric model needs further investigation, otherwise the heuristic of denoising features might not be applicable.
2. More explanation is needed on the insights of where the rationale of denoising comes from, e.g. lines 159-190.
3. Since the eigendecomposition is applied on adjacency, the comparison of training and inference time costs between the vanilla GNN and with the add-on of the proposal is needed.

**Questions:**

1. In Figure 2, why do the two subgraphs compare w.r.t. amplitude and normalized amplitude, respectively?
2. Regarding section 2.2, can you explain more about how the structure is revealed parametrically? In line 137, $|\lambda|$ is considered the signal-to-noise ratio, when it can also be considered the degree of homophility?
3. The remark below definition 1, ''the information about the labels is indeed contained in the leading vectors of V and U'', is unclear.Because usually the decomposition of a matrix gives a support space, the weights of each support, and the coefficient of reconstructing the matrix using the support, which leads to the insufficiency of the leading support vectors to be representative.
This should be relevant for W2.

---

> ### Author Response · Authors · 2024-11-19
> **Reply to Reviewer Pd13 (Part 1/2)**
>
> We thank the reviewer for thoughtful comments, positive review and the interesting questions, especially for drawing our attention to the derivation of JDR via the cSBM and its implications. In the following, we address the reviewer's concerns one by one.
>
> **Weaknesses:**
>
> > 1. The theoretical framework seems convincing to me, while how the real data sets fit the parametric model needs further investigation, otherwise the heuristic of denoising features might not be applicable.
>
> This is indeed an important topic and we discuss it in Appendix A.2, where we demonstrate the significance of the cSBM in GNN research. We wish to highlight two points: First, our extensive experiments clearly confirm that the reasoning for cSBM does transfer to real-world datasets to a large extent. Second, there is substantial evidence in the literature that the cSBM is a useful model in the analysis of GNNs and graph datasets. Important works include the theoretical studies of node-level GNN problems based on the cSBM, e.g., on double descent (Shi et al., 2024), neural collapse (Kothapalli et al., 2023), OOD generalization (Baranwal et al., 2021), or oversmoothing (Wu et al., 2023). Beyond being a standard synthetic benchmark, the cSBM is also used to verify hypotheses about GNNs (Ma et al., 2022; Luan et al., 2023).
> This does not mean that all the work is done. Like any model, the cSBM idealizes certain aspects of graph data which in some cases is a limitation; we discuss this as well in Appendix A.2. But there is overwhelming evidence (in particular experimental) that it is a very useful model to reason about graph data.
>
> > 2. More explanation is needed on the insights of where the rationale of denoising comes from, e.g. lines 159-190.
>
> Thank you for bringing this lack of clarity to our attention. We will improve these explanations in the revised manuscript along the following lines:
> Due to the quadratic and bilinear forms in (2), the optimal $\mathbf{v}^*$ is related to the leading singular vector in $X$ and second leading eigenvector in $A$. The reason for this is that each of the two subproblems would lead exactly to the corresponding eigen/singular vectors, so the overall formulation leads to a compromise between the two.  Too see this, for simplicity, let us only consider $A$. In a balanced two-class SBM, we can write the adjacency as
> $$
> A = \mathbb{E}(A) + (A - \mathbb{E}(A))
> $$
> where $\mathbb{E}(A)$ captures the expected structure of $A$ and $(A - \mathbb{E}(A))$ the fluctuations around its mean.
> Now consider without loss of generality that the nodes are sorted such that the first half belongs to class $1$ and the second half to class $2$. Then $\mathbb{E}(A)$ is a block matrix with structure
>
> $$
> \mathbb{E}(A) =
> \begin{bmatrix}
> p \mathbf{1}\mathbf{1}^T  & q\mathbf{1}\mathbf{1}^T  \newline
> q\mathbf{1}\mathbf{1}^T & p\mathbf{1}\mathbf{1}^T
> \end{bmatrix}
> $$
> where $p$ is the probability of links between nodes of the same class and $q$ between nodes of different classes. Let the vector of class labels be $\mathbf{y} = [1, \ldots, 1, -1, \ldots, -1]^T$. Then we can write
> $$
> \mathbb{E}(A)=\frac{p+q}{2}\mathbf{1}\mathbf{1}^T+\frac{p-q}{2}\mathbf{y}\mathbf{y}^T.
> $$
> For small fluctuations (high SNR, $|\lambda| \gg 1$), $A \approx \mathbb{E}(A)$. Then $A$ is approximately a rank-$2$ matrix and we can directly see that $\mathbf{y}$ is the second leading eigenvector. Even for lower SNRs there will be substantial alignment between $\mathbf{y}$ and the label vector. A similar reasoning can be applied for the leading left singular vector in $X$. Together this shows that the optimal $\mathbf{v}^*$ is related to the labels $\mathbf{y}$. We hope that this provides more clarity; if not we would be happy to answer further questions.
>
> > 3. Since the eigendecomposition is applied on adjacency, the comparison of training and inference time costs between the vanilla GNN and with the add-on of the proposal is needed.
>
> We thank the reviewer for the suggestion (together with FthW who made similar remarks). We agree that this comparison would strengthen the paper. We ran the requested experiments and added them to the revised paper. We also discuss implications and limitations of such an analysis. The results do not show a clear "winner"; JDR generally requires a comparable or shorter time compared to baselines. The reasons for these ambiguities are, among others, that results  depend on many hardware and framework-related factors. In our particular case we should add that different hyperparameter choices of the rewiring methods lead to dramatically different run times (even when applying the same method on the same dataset). For example, FoSR is quite fast on the large Twitch-gamers dataset as it only runs for $5$ iterations, while it is slow on the smaller Questions dataset, where it runs for $700$ iterations.

---

> > ### Author Response · Authors · 2024-11-19
> > **Reply to Reviewer Pd13 (Part 2/2)**
> >
> > **Questions:**
> >
> > > 1. In Figure 2, why do the two subgraphs compare w.r.t. amplitude and normalized amplitude, respectively?
> >
> > This is indeed a typo; we corrected it to be "amplitude" in both cases.
> >
> > > 2. Regarding section 2.2, can you explain more about how the structure is revealed parametrically? In line 137, $|\lambda|$ is considered the signal-to-noise ratio, when it can also be considered the degree of homophility?
> >
> > For the first part of the questions, see our answer to your W.2. For the second part, this is an intrinsic property of the cSBM, which assumes that "homophily" is linear in the SNR $\lambda$. In general, $|\lambda|$ characterizes the graph SNR, while the sign together with $|\lambda|$ defines the degree of hetero- or homophily. A graph can have a high SNR regardless of whether it is heterophilic or homophilic. This can be verified by considering the block matrix representation of $\mathbb{E}(A)$ from our answer to W.2. $p$ and $q$ are directly related to $\lambda$ via the average node degree, so consider $p = 1$ and $q=0$ meaning that there can be only in-class connections. In this case the adjacency matrix is a block-diagonal matrix without fluctuations, so obviously high (perfect) SNR and also clearly homophilic. The same argument can be made for $p = 0$ and $q=1$ which is heterophilic and again has perfect SNR. But we should point out that while interpreting $|\lambda|$ as a measure of homophily (or heterophily) is standard, there is no definitive agreement on the right way to measure it, and there are many suggestions in the literature.
> >
> > > 3. The remark below definition 1, ''the information about the labels is indeed contained in the leading vectors of V and U'', is unclear.Because usually the decomposition of a matrix gives a support space, the weights of each support, and the coefficient of reconstructing the matrix using the support, which leads to the insufficiency of the leading support vectors to be representative. This should be relevant for W2.
> >
> > Thanks for pointing this out. We will clarify it in the revised paper. As you already suggest this is related to your W.2. For a general matrix with full rank, it is of course not the case that only the leading vectors contain the information, as is the standard argument for truncated SVD. The case here is different in that we are still assuming a cSBM-like model, so in the high SNR case, the matrices are approximately low rank. To see this, we can simply extend W.2 to several ($k$) communities. Here, too, a block matrix representation is now given with several blocks and it can be shown by similar arguments as above that the class assignment is contained in the $k$ leading eigenvectors. This is also related to the SNR; if the signal is too weak (below the detection threshold), class information is not only contained in the leading eigenvectors, but will be distributed over the entire support space, as you suggest in the question. For deeper insight into multi-class SBM, we recommend the [monograph](https://jmlr.org/papers/v18/16-480.html) of Emmanuel Abbe from 2018.
> >
> >
> > We hope this addresses your questions and look forward to further feedback.

---

> > > ### Comment · Reviewer_Pd13 · 2024-11-21
> > >
> > > I'd like to thank all the authors for such a good rebuttal. I think all my concerns have been addressed by the authors' feedback. And I also thank the authors for their great attitude in answering all my questions. So I'm raising my score to 8. Best of luck.

---

### Official Review · Reviewer_PNCZ · 2024-10-31

**Soundness:** 3
**Presentation:** 3
**Contribution:** 2
**Rating:** 8
**Confidence:** 3

**Summary:**

This paper considers the graph rewiring problem and feature denoising problem to improve the graph node classification task.
This paper utilizes cSBM modeling to optimize both graph structures and graph features to make them have better alignment, then proposes the JDR method to effectively optimize the alignment function.

**Strengths:**

- This paper uses cSBMs as a key framework to build intuition about the graph rewiring and denoising problem, providing the theoretical foundation for the alignment target.
- The empirical verification using synthetic data is clear.
- The method is evaluated on both homophilic graphs and heterophilic graphs, showing its generalizability.

**Weaknesses:**

- The proposed method involves graph structure matrix and graph feature matrix decomposition, which can be computationally challenging on extremely large real-world graph data, limiting the practicability of the proposed method.
- As the SVD decomposition can have time complexity of $O(N^3)$, it may be not accurate to say the proposed JDR has time complexity of $O(N)$

**Questions:**

- As the computation overhead of the proposed method seems large, can it be applied to large graphs, e.g. ogbn-products?

---

> ### Author Response · Authors · 2024-11-19
> **Reply to Reviewer PNCZ**
>
> We thank the reviewer for their positive comments and thoughts and suggestions on computational aspects of JDR. In the following, we  address the reviewer's concerns. A revised manuscript will be uploaded soon. We look forward to further feedback.
>
> **Weaknesses:**
>
> > The proposed method involves graph structure matrix and graph feature matrix decomposition, which can be computationally challenging on extremely large real-world graph data, limiting the practicability of the proposed method.
>
> As you suggest, it may be challenging to apply JDR out of the box to *extremely* large graphs. But this is true for any method, and in particular for all preprocessing rewiring methods. We specifically discuss the issue of scalability in Section 3.2 of the main paper. While scalability is a general limitation of preprocessing rewiring methods (if one wants to work with huge graphs), JDR is at least as good and in many cases more scalable than the baselines. We show in the accompanying experiments that we can still use JDR on graphs up to size of $168$k nodes and millions of edges and outperform the baseline methods. The strongest baselines like BORF are not applicable in this regime. JDR handles these cases without specific optimization for very large graphs. By carefully engineering a more efficient implementation, we are confident that JDR can be extended to graphs with millions of nodes (as for the ogbn-products mentioned below), but that entails substantial new research which is outside the focus of this paper.
>
>
> > As the SVD decomposition can have time complexity of $O(N^3)$, it may be not accurate to say the proposed JDR has time complexity of $O(N)$
>
> Thank you for pointing this out; we should state it more clearly in the revised version. Our estimate is based on the following reasoning: 1) real graphs are more or less sparse, and 2) we only need the top-$k$ spectral spaces, not the full eigen/singular value decompositions. We discuss this in Appendix A.1.3:
> The complexity to obtain the top-$k$ eigenvectors (using variants of the power method) scales as $\mathcal{O}(N^2k)$. For sparse matrices this further reduces to $\mathcal{O}(\text{nnz}(A)k)$ where $\text{nnz}(A)$ is the number of non-zero elements in $A$. If the average degree $d$ does not scale with $N$, the complexity becomes $\mathcal{O}(N)$. As mentioned above, we will state these estimates much more explicitly in the revised manuscript.
>
>
> **Questions:**
>
> > As the computation overhead of the proposed method seems large, can it be applied to large graphs, e.g. ogbn-products?
>
> Please refer to our response to W.1. In a nutshell: graphs with more than $200$k nodes would present a challenge for our current code, but we believe a careful implementation of sparsity-aware power method would be able to handle them.

---

> > ### Comment · Reviewer_PNCZ · 2024-11-25
> > **response**
> >
> > Thank authors for the clarification and i raise my score from 6 to 8.

---

### Official Review · Reviewer_VArZ · 2024-11-01

**Soundness:** 3
**Presentation:** 3
**Contribution:** 3
**Rating:** 8
**Confidence:** 2

**Summary:**

The paper proposes to  jointly consider graph rewiring and feature denoising to improve the performance of GNNs for downstream classification tasks.  The proposed method improves the performance of a numer of GNNs on a number of popular datasets in the experiments.

**Strengths:**

1. The proposed method is general and can be applied to a wide range of GNNs for downstream classification tasks.

2. The proposed method jointly considers graph rewiring and feature denoising.

3. The proposed method improves the performance of a numer of GNNs on a number of popular datasets in the experiments.

**Weaknesses:**

1. The solution is somewhat incremental and its novelty is low, although it appears to be sound.

2. There is no theoretical guarantee on the degree of improvement using JDR.

3. The experiments were conducted on a small number of datasets that cannot be considered as an evidence that the proposed method is really effective.

**Questions:**

My main concern with this work is that although the proposed method looks to be sound, there is really no theorectical ground on its effectiveness in general when applied to a wider range of datasets. Also, the datasets used are very small and I'm not sure how useful and how scalable the method is in practice. It would significantly strengthen the work if the authors can either theoretically show how much improvement can be obtained using their method or empirically demonstration the effectiveness on a wide range of datasets, not just on common datasets that we have already seen too many papers claiming their effectiveness over others. I believe the contributions of such work are rather limited.

---

> ### Author Response · Authors · 2024-11-15
> **Reply to Reviewer VArZ**
>
> **Weaknesses:**
>
> 1. We disagree with this statement. Can you elaborate in what sense or relative to what earlier work is our manuscript somewhat incremental or of low novelty?
> We don't know of other methods that rewire the graph with feature information (and even less so of methods that jointly denoise both). We believe that our contribution is rather original. We also think that our approach to analysis is original (certainly not incremental) and we are confident that it will inspire new theoretical work. We have not seen anything related in the literature.
>
> 2. It would indeed be great to have a guarantee for downstream performance
> on node classification problems, but showing something like this is extremely challenging as it depends on the downstream GNNs and unknown data distributions. Such guarantees are also not available for any other rewiring method.
> In Proposition 1, we proved that JDR increases alignment of graph and node features with the labels under the cSBM assumption. We agree that there is still a gap between this and quantifying the degree of improvement, but at the same time we emphasize that the proof we have provided is the first of its kind and a step in this direction. We are working on a broader quantitative theory, but this will be a contribution on its own as it is non-trivial even under the assumption of synthetic data and a single-layer linear GCN. We do provide extensive numerical experiments and ablations which demonstrate effectiveness of JDR with overwhelming statistical significance.
>
> 3. We strongly disagree with the reviewer on this statement. The amount of experimental evaluation in our work is almost excessive. We have evaluated the effectiveness of JDR on numerous datasets and in numerous experiments and combinations. This was explicitly pointed out by reviewer Fauu end implied by others. Here are concrete numbers: In the main text we provide results on **17 synthetic** and **13 real-world** datasets with different sizes, properties and homophily levels. This results in **214 distinct experiments**. In the appendix, we further extend these with three additional real-world datasets, additional splits, ablations on JDR(X) and JDR(A), ablations on the parameters of JDR and experiments with spectral clustering **resulting in additional 300+ experiments**.
> Comparing this to related papers on FoSR or BORF, we not only consider many more datasets, but also provide much more diverse experiments. We respectfully ask the reviewer to reconsider their judgment in the light of these facts.
>
> **Questions:**
> > My main concern with this work is [...] there is really no theorectical ground on its effectiveness in general when applied to a wider range of datasets.
>
> Please refer to our response to W.2 and W.3.
>
> > Also, the datasets used are very small and I'm not sure how useful and how scalable the method is in practice.
>
> We specifically discuss scalability in Section 3.2 of the main paper. While scalability is a general limitation of preprocessing rewiring methods (if one wants to work with huge graphs), JDR is at least as good and in many cases more scalable than the baselines. We show in the accompanying experiments that we can still use JDR on  graphs up to size of 168k nodes and millions of edges and outperform the baseline methods. The strongest baseline methods like BORF are not applicable in this regime. JDR handles these cases without specific optimization for very large graphs. By carefully engineering a more efficient implementation, we are confident that JDR can be extended to graphs with millions of nodes, but that entails substantial new research which is outside the focus of this paper. The original theoretical guarantees we give are a good indication of SNR regimes where JDR can be effective regardless of graph size. Most real graphs from biology and medicine that we deal with on a daily basis have numbers of vertices in the thousands, occasionally tens of thousands, suggesting that these sizes are relevant for applications.
>
> > [...] if the authors can either theoretically show how much improvement can be obtained using their method
>
> Please see our response to W.2.
>
> > or empirically demonstration the effectiveness on a wide range of datasets, not just on common datasets [...]
>
> Some of the datasets we used are indeed established benchmarks. This is the norm in the community and it enables rigorous comparison with baselines which is invaluable for fair evaluation. That said, as we explained above, we used an enormous diversity of datasets, including some that are rarely seen in the rewiring literature (e.g. Platonov et al., 2023 and Lim et al., 2021.) To our knowledge, the datasets we compiled for this paper never appeared together in a single publication in the field. Again, we respectfully ask that you reconsider your judgment in the light of the diversity of experiments we performed.

---

> > ### Comment · Reviewer_VArZ · 2024-11-25
> >
> > The rebuttal has addressed my concerns and I'd like to raise my rating to 8. Thank you for the clarification.

---

### Official Review · Reviewer_FthW · 2024-11-04

**Soundness:** 4
**Presentation:** 4
**Contribution:** 4
**Rating:** 8
**Confidence:** 3

**Summary:**

The paper presents Joint Denoising and Rewiring (JDR), a novel algorithm designed to simultaneously address noisy graph structures and node features, thereby enhancing node classification performance in graph neural networks (GNNs). The work proposed aims to address the issue of graph datasets that suffer from noise due to missing or spurious edges and misaligned node features. JDR counters this by iteratively refining both the graph structure and features to maximize the alignment between their leading eigenspaces, achieving what the authors call “spectral resonance,” which improves data representation for downstream GNN performance. The algorithm operates in three iterative steps: first, it decomposes the graph’s adjacency matrix and node feature matrix into eigen and singular vectors; then, it aligns the leading eigenvectors of the graph with the primary singular vectors of the feature matrix, creating a more cohesive representation; and finally, it synthesizes the updated graph for subsequent GNN tasks. The authors argue this iterative approach effectively enhances graph quality and feature clarity, especially in datasets exhibiting mixed homophily and heterophily, which are often challenging for GNNs. The proposed work provides extensive experiments show that JDR consistently outperforms existing rewiring methods on synthetic and real-world datasets, yielding better alignment and improved node classification accuracy. Additionally, JDR’s preprocessing method produces an enhanced graph that supports interpretability and reusability, unlike other methods that modify graphs solely during training. The paper highlights JDR’s scalability and versatility across different graph types, underscoring its potential in handling noisy real-world data and establishing it as a valuable advancement in preprocessing techniques for GNNs.

**Strengths:**

The primary novelty of JDR lies in its combined optimization of graph structure and node feature alignment, enhancing data quality by maximizing alignment between the spectral components of the graph and feature matrices. This unified approach addresses both structural and feature-level noise simultaneously, which is rare among existing methods that typically target these types of noise separately. A key concept introduced is “spectral resonance,” where optimal alignment between the graph’s leading eigenvectors and the feature matrix’s singular vectors is achieved, providing a measurable target for denoising and boosting node classification performance. To manage the challenging non-convex optimization, the paper presents an iterative heuristic based on alternating optimization, which simplifies alignment maximization and enables efficient processing of large, real-world graph datasets with multiple classes. Another advantage of JDR is that it outputs a modified graph in the preprocessing stage, enhancing interpretability and reusability for subsequent GNN applications—a contrast to end-to-end methods that alter the graph only during training. Lastly, JDR’s design allows it to adapt effectively to both homophilic and heterophilic graphs, expanding its applicability beyond previous methods, which often work best with specific types of graph structures, such as those with high homophily.

The paper is well-organized, systematically guiding the reader through the challenges, methodology, and outcomes of the proposed Joint Denoising and Rewiring (JDR) algorithm. It begins with an introduction that effectively frames the problem of noisy graph structures and features, establishing the need for an approach that addresses both in unison. The methodology section details the concept of spectral resonance and the iterative optimization heuristic that drives JDR, using clear mathematical definitions and visual aids to support understanding. Following this, a comprehensive experimental section validates the algorithm's effectiveness across synthetic and real-world datasets, offering detailed comparisons with state-of-the-art methods and highlighting the algorithm’s robustness across homophilic and heterophilic graph types. Finally, the paper provides an insightful discussion on related works, situating JDR within the broader landscape of graph preprocessing techniques, before concluding with a summary of contributions and potential directions for future research. Overall, the structure flows logically, with each section building on the previous one to reinforce the practical relevance and theoretical underpinnings of JDR.

Addressing the limitations of the proposed approach is appreciated

The illustrations provided in the work, both in the body and abstract, are informative and well done as well as qualitatively informative.

 The use of the appendix is also well done in providing explicit discussion of the proposed algorithm

The paper has a robust experimental section with compelling results working in favor of the approaches proposed

**Weaknesses:**

JDR depends on the availability of informative node features for effective rewiring and denoising, which restricts its applicability to settings with substantial node feature information; this reliance could limit its effectiveness in networks that primarily encode structural data. The algorithm’s design is also tailored to node-level tasks, making it less suited for graph-level tasks like graph classification, where global structure matters more than node-specific features

**Questions:**

could timing comparisons of the proposed methods, more explicitly and beyond just complexity comparison, be provided to observe computational benefits/limitations?

---

> ### Author Response · Authors · 2024-11-19
> **Reply to Reviewer FthW**
>
> We thank the reviewer for their positive comments and feedback. In the following we address the concerns mentioned by the reviewer. We will upload an accordingly revised version of the paper very soon.
>
> **Weaknesses:**
>
> > JDR depends on the availability of informative node features for effective rewiring and denoising, which restricts its applicability to settings with substantial node feature information; this reliance could limit its effectiveness in networks that primarily encode structural data.
>
> This is indeed true. We point it out in the paper and discuss it as an intrinsic limitation of our approach. At the same time, we emphasize that it is also an advantage of our method over others that it can leverage the information contained in the node features if they are available.
>
> > The algorithm’s design is also tailored to node-level tasks, making it less suited for graph-level tasks like graph classification, where global structure matters more than node-specific features
>
> We agree with this as well. (We discuss it as well in the paper.) The reason for this is that the derivation of JDR is closely related to the cSBM and thus also adopts some of its limitations. How to extend the cSBM idea to graph-level problems is an interesting fundamental research question.
>
> **Questions:**
>
> > could timing comparisons of the proposed methods, more explicitly and beyond just complexity comparison, be provided to observe computational benefits/limitations?
>
> This is indeed possible (reviewer Pd13 made a similar remark). We ran the requested experiments and added them to the revised paper. We also discuss implications and limitations of such an analysis. The results do not show a clear "winner"; JDR generally requires a comparable or shorter time compared to baselines. The reasons for these ambiguities are, among others, that results  depend on many hardware and framework-related factors. In our particular case we should add that different hyperparameter choices of the rewiring methods lead to dramatically different run times (even when applying the same method on the same dataset). For example, FoSR is quite fast on the large Twitch-gamers dataset as it only runs for $5$ iterations, while it is slow on the smaller Questions dataset, where it runs for $700$ iterations.

---

### Author Response · Authors · 2024-11-21
**Paper revision and change log**

We thank the reviewers for thoughtful feedback and for recognizing that the proposed method is novel and interesting (FthW, Pd13, Fauu), that the experiments are comprehensive and convincing (FthW, PNCZ, Pd13, Fauu), and that the prose is clear and visualizations are helpful and informative.

We appreciate the suggestions to include additional experimental runtime ablations and comparisons between MLPs and GNNs; we ran the requested experiments and include the results in the revised manuscript. We believe that these further strengthen our empirical evaluation.

Attached you find a revised version of our paper. For easy reference, we color-coded all changes in blue.
We summarize the changes below. Point-by-point answers are included as a response to each reviewer.

**Empirical Additions**
- Runtime comparisons: Added in Table 5, Appendix A.1.4 (FthW, Pd13).
- MLP and JDR experiments: Added in Tables 10, 11, and Figure 6, Appendix A.5.3 (Fauu).

**Discussion Improvements**
- Expanded discussion on computational complexity of JDR (PNCZ).
- Added more explanation on the denoising rationale (Pd13).
- Reformulated the limitations of JDR on node features (FthW, Fauu).

**Clarity Improvements**
- Corrected y-label in Figure 2 (Pd13).
- Clarified Remark after Definition 1 (Pd13).
- Reworked introduction of Proposition 1 in Section 2.3 and proof in Appendix A.3 (Fauu).
- Relocated and expanded discussion of Dong and Kluger (2023) under graph denoising in related work (Fauu).

Finally, we want to emphasize that although we have mainly received feedback on the empirical evaluation of our method, our work provides novel theoretical contributions (as acknowledged by Pd13 and Fauu): Under a stylized cSBM-like model, JDR's interpolation provably improves alignment of the largest eigenvector of $A$ and $X$ with the labels $\mathbf{y}$ for sufficiently large graphs. This original result is based on the BBP transition in the spiked covariance model (Baik et al., 2005) and eigenvalue perturbation methods, detailed in Appendix A.3. We believe that this first theoretical results will motivate follow-up work, both theoretical and practical.

We thank the reviewers again for their helpful feedback and welcome any additional comments or suggestions that can help to further improve the paper.

---

### Meta-Review · Area_Chair_ZfkD · 2024-12-20

**Metareview:**

This paper is a clear accept -- the reviewers found the presented method to be novel, interesting, and well-presented. The idea of aligning the singular vector subspaces of node features and the graph adjacency matrix as a way of jointly denoising feature data and rewiring the graph is intuitive and seemingly effective.

The theoretical results, while holding in a highly simplified setting (cSBMs) help build intuition for/justification of the method. The empirical results are strong and comprehensive. During the rebuttal the authors effectively addressed several initial concerns of the reviewers, and the reviewers raised their scores accordingly.

**Additional Comments On Reviewer Discussion:**

See main metareview. Given the positive reviews of the paper, the rebuttal, while helpful in clarifying some points, did not end up impacting the paper decision.

---

### Decision · Program_Chairs · 2025-01-22

Accept (Oral)